# Strong, tough, rapid-recovery, and fatigue-resistant hydrogels made of picot peptide fibres

Bin Xue [1,2,6], Zoobia Bashir[1,6], Yachong Guo[3], Wenting Yu[1], Wenxu Sun[1], Yiran Li [1], Yiyang Zhang[1], Meng Qin[1], Wei Wang [1,4] ✉ & Yi Cao [1,2,4,5] ✉

Hydrogels are promising soft materials as tissue engineering scaffolds, stretchable sensors, and soft robotics. Yet, it remains challenging to develop synthetic hydrogels with mechanical stability and durability similar to those of the connective tissues. Many of the necessary mechanical properties, such as high strength, high toughness, rapid recovery, and high fatigue resistance, generally cannot be established together using conventional polymer networks. Here we present a type of hydrogels comprising hierarchical structures of picot fibres made of copper-bound self-assembling peptide strands with zipped flexible hidden length. The redundant hidden lengths allow the fibres to be extended to dissipate mechanical load without reducing network connectivity, making the hydrogels robust against damage. The hydrogels possess high strength, good toughness, high fatigue threshold, and rapid recovery, comparable to or even outperforming those of articular cartilage. Our study highlights the unique possibility of tailoring hydrogel network structures at the molecular level to improve their mechanical performance.

Many natural load-bearing tissues can restore their baseline mechanical properties quickly and endure many mechanical cycles (in some cases, many per day over many years), indicating remarkable mechanical stability and reliability. For example, our articular cartilage can be compressed by more than 40% strain and forces six times the human body weight at a frequency of 0.5 Hz without showing noticeable mechanical fatigue[1–4]. Inspired by the amazing mechanical properties of load-bearing tissues, numerous efforts have been devoted to engineering hydrogels that can function under intensive mechanical loads for applications such as musculoskeletal repairing, cartilage regeneration and soft robot construction[5–7]. A key design principle used to toughen hydrogels is to dissipate energy using sacrificial bonds/networks[8,9], which may rely on hydrophobic interactions[10–12], ionic pairing[13–15], hydrogen bonding[16,17], coordination interactions[18,19], host-guest interactions[20] and microcrystals[21,22]. Despite substantially

toughening hydrogels, these molecular engineering approaches often fail to strengthen the hydrogels at the same time, because strength and toughness are mutually exclusive[23–25]. Moreover, although these weak interactions are dynamic and reversible at the molecular level, the hydrogels show slow and limited recovery after deformation because of the huge entropy cost of reforming these dynamic bonds in the hydrogel networks[26–29]. As these sacrificial bonds cannot reform efficiently, the hydrogels lack a mechanism to prevent crack propagation and are susceptible to fatigue under cyclic loading[30]. Recently, some efforts have partially overcome the trade-offs among these mechanical properties using strain-induced self-reinforcement[31], chain entanglement[32] or mechanochemically activable hidden length[33]. For example, Liu et al. used the damageless reinforcement strategy of strain-induced crystallisation to build high toughness and rapid recovery[31]. Kim et al. used dense entanglements, which greatly

[1]Collaborative Innovation Center of Advanced Microstructures, National Laboratory of Solid State Microstructure, Department of Physics, Nanjing University, Nanjing 210093, China. [2]Jinan Microecological Biomedicine Shandong Laboratory, Jinan, China. [3]Kuang Yaming Honors School, Nanjing University, Nanjing 210023, China. [4]Institute for Brain Sciences, Nanjing University, Nanjing 210093, China. [5]Chemistry and Biomedicine innovation center, Nanjing University, Nanjing 210093, China. [6]These authors contributed equally: Bin Xue, Zoobia Bashir. ✉e-mail: wangwei@nju.edu.cn; caoyi@nju.edu.cn

outnumber crosslinks in hydrogels, to realise high toughness, strength, and fatigue resistance[32]. Moreover, the hidden length in fibril structures or folded proteins can provide extra extensibility after being released, thus enhancing the toughness of hydrogels[33–36]. However, it remains challenging to achieve high strength, high toughness, robust fatigue resistance, and rapid recovery in the same synthetic hydrogel.

In nature, load-bearing tissues such as muscle, tendon, and cartilage possess complex hierarchical structures spanning different length scales[4,37,38], which may represent an approach for designing synthetic hydrogels with excellent mechanical properties. As recently demonstrated independently by the He group[39] and Liu group[40], introducing hierarchical structures into hydrogels by freeze-casting and additional processing can be an effective way of improving both strength and toughness. Although these approaches can strengthen and toughen hydrogels, they inevitably make the hydrogels inhomogeneous and anisotropic. In natural load-bearing materials, hierarchical structures are formed through complicated self-assembly[41]. Through millions of years of evolution, nanometre-scale structures and interactions at molecular levels are evolved to integrate high strength and toughness, rapid recovery, and anti-fatigue properties[19]. A hallmark of biological tissue networks, such as tendon, muscle and bone, is the combination of self-assembled fibrous nanostructures and abundant non-fibrillar organic matrix to efficiently mediate load distribution and energy dissipation[42].

Inspired by the structures of biological networks, we propose a type of hierarchical structure made of picot fibres (p-fibres) consisting of self-assembled metal ion-clad peptide β-strands interconnected by flexible hidden lengths. By crosslinking these fibres, the resulting hydrogels showed an unusual combination of high mechanical strength (fracture stress ~4.1 MPa), ultrahigh toughness (fracture energy ~25.3 kJ m$^{-2}$), excellent fatigue resistance (fatigue threshold ~424 J m$^{-2}$) and almost 100% mechanical recovery in one second. This design provides a general route to combining typically incompatible mechanical properties in synthetic hydrogels.

## Results and discussion
### Design of hydrogels based on metal ion-clad picot fibres
In conventional double-network hydrogels, the primary tightly crosslinked network is responsible for energy dissipation, and the secondary loosely crosslinked network provides hidden length (Fig. 1a). When the tight network is crosslinked by physical interactions, in principle, the hydrogel can recover its baseline mechanical properties after the load is released. However, in the recovery process, restoration of the original gel shape is driven solely by the recoiling of the secondary loose network, and the reformation of the tight network is entropically unfavourable. Here, we propose a type of hydrogel network made of picot fibres, in which the hidden length and ability to dissipate energy are both integrated into the picot fibres (Fig. 1b). The picot fibres are made of long flexible polymers decorated with self-assembling peptides. Driven by the self-assembly of the peptides, the links between peptides are zipped as picots to contribute to the hidden length (Fig. 1b). The picots are at different lengths and the peptides connected with each other randomly. Rupturing the strong inter-peptide interactions, such as hydrogen bonding in the picot fibres efficiently dissipates energy and releases the hidden length without reducing the network connectivity of the hydrogels. Thus, this design resolves the conflicts of strength and toughness in common polymer networks. Moreover, reformation of the individual picot fibres occurs locally and independently, which ensures rapid recovery of the hydrogel structures.

We took inspiration from the strong and tough mussel fibres and used the histidine-rich motif of mussel foot protein Mfp-4[37], Gly-His-Val-His-Thr-His-Arg-Val-Leu-His-Lys (denoted GK$_{11}$), to construct the picot fibres (Fig. 1c and Supplementary Fig. 1a–c). GK$_{11}$ fibres were strengthened upon Cu$^{2+}$ binding to yield metal ion-clad picot fibres. To incorporate hidden length in the peptide fibres, we introduced an acylate group to the N-terminus of GK$_{11}$ (named ACLT-GK$_{11}$, Supplementary Fig. 1d) and polymerised this peptide with acrylamide (Fig. 1c). Using Alphafold2[43], we predicted that the structure of the metal ion-clad peptide fibres forms well-aligned anti-parallel β-sheets connected by six pairs of hydrogen bonds (Fig. 1d). Cu$^{2+}$ preferentially coordinates with three histidine residues in the N-terminal of a peptide (A: H2, A: H4 and A: H6) and one residue in the C-terminal of the neighbouring peptide (B: H10). The binding energy between the GK$_{11}$ peptide and Cu$^{2+}$ was −7.2 kcal mol$^{-1}$ as determined by simulation, which is about five times the strength of a typical hydrogen bond in proteins[44,45]. Noting that the β-sheet of a single GK$_{11}$ peptide is slightly twisted (θ ~12°), making side chains of histidine (imidazole) distributing on both sides of the β-sheet. Considering the tetrahedral coordination of Cu$^{2+}$ and histidine, the second Cu$^{2+}$ prefers to form coordination on the other side of the β-sheet with a third strand.

### Characterisation of supramolecular fibres and picot fibres
GK$_{11}$ peptides can self-assemble into supramolecular fibres and bound with Cu$^{2+}$ ions. The supramolecular peptide fibres without and with ions were denoted as GK$_{11}$ and GK$_{11}$/Cu$^{2+}$ fibres, respectively. Atomic force microscopy (AFM) images (Fig. 2a) indicated that the diameters of both GK$_{11}$ and GK$_{11}$/Cu$^{2+}$ fibres were ~1–2 nm. Without Cu$^{2+}$, the peptide fibres were straight, but with Cu$^{2+}$, the fibres were curved and prone to form bundles, branches, and entanglements. We used circular dichroism (CD) and Fourier transform infrared (FT-IR) analyses to investigate the secondary structures of GK$_{11}$ in the fibres (Fig. 2b, c). The CD peak in the range of 214–220 nm indicated the presence of β-sheet structures of the peptide fibres (Fig. 2b)[46]. Moreover, upon Cu$^{2+}$ binding, an additional FT-IR peak at ~1622 cm$^{-1}$ appeared in the FT-IR spectrum in addition to the absorption peak at ~1672 cm$^{-1}$ (Fig. 2c), which may imply that the β-sheet structures became twisted[47]. We then determined the binding stoichiometry of Cu$^{2+}$ to GK$_{11}$ by ultraviolet-visible (UV–vis) spectroscopy and X-ray fluorescence spectrometry (XRFS) (Supplementary Figs. 2, 3a). The binding ratio of Cu$^{2+}$ and GK$_{11}$ was ~1.0, consistent with the stoichiometry of the predicted fibre structures (Fig. 1d). In addition, the melting temperature of GK$_{11}$ increased by ~11 °C upon Cu$^{2+}$ binding (Supplementary Fig. 3b), which confirms the presence of strong interactions between GK$_{11}$ and Cu$^{2+}$.

Next, we investigated the nanostructure of picot fibres and metal ion-clad picot fibres. The picot GK$_{11}$ fibres without Cu$^{2+}$ (pGK$_{11}$) were prepared by copolymerisation of acrylamide and ACLT-GK$_{11}$ in deionized water and dialysed the polymers in Tris buffer to induce self-assembly. The Cu$^{2+}$-clad picot GK$_{11}$ fibres (pGK$_{11}$/Cu$^{2+}$) were prepared under the same conditions except that they were dialysed in Tris buffer containing 200 mM CuCl$_2$. AFM images indicate that the picot fibres with and without Cu$^{2+}$ formed thicker bundled nanofibres (~2.0 nm for pGK$_{11}$ and ~2.2 nm for pGK$_{11}$/Cu$^{2+}$, Fig. 2d) than the supramolecular fibres (Fig. 2a). Moreover, advanced microscale hierarchical structures, such as coiled picot fibres, were observed due to entanglement of the unstructured polymer chains. According to the results of AFM-based nanoindentation measurements, the picot fibres were less stiff than the fibres formed without acrylamide (Supplementary Fig. 4), indicating the successful incorporation of hidden length into the peptide fibres.

Furthermore, the hidden length in picot fibres can be tuned by adjusting the acrylamide concentration. As shown in Supplementary Fig. 5, the metal ion-clad picot fibres became thicker at the higher ratio of acrylamide and peptide, indicating that the increasing ratio of polyacrylamide led to a long hidden length. This was also confirmed by the AFM-based single molecule force spectroscopy (SMFS) of metal ion-clad picot fibres (Supplementary Fig. 6). The hidden length increased from ~5.4 to ~9.5 nm when the molar ratio of acrylamide and ACLT-GK$_{11}$ increased from 16.3 to 32.6. The measured values of the hidden length were close to theoretical values of 4.9 and 9.8 nm,

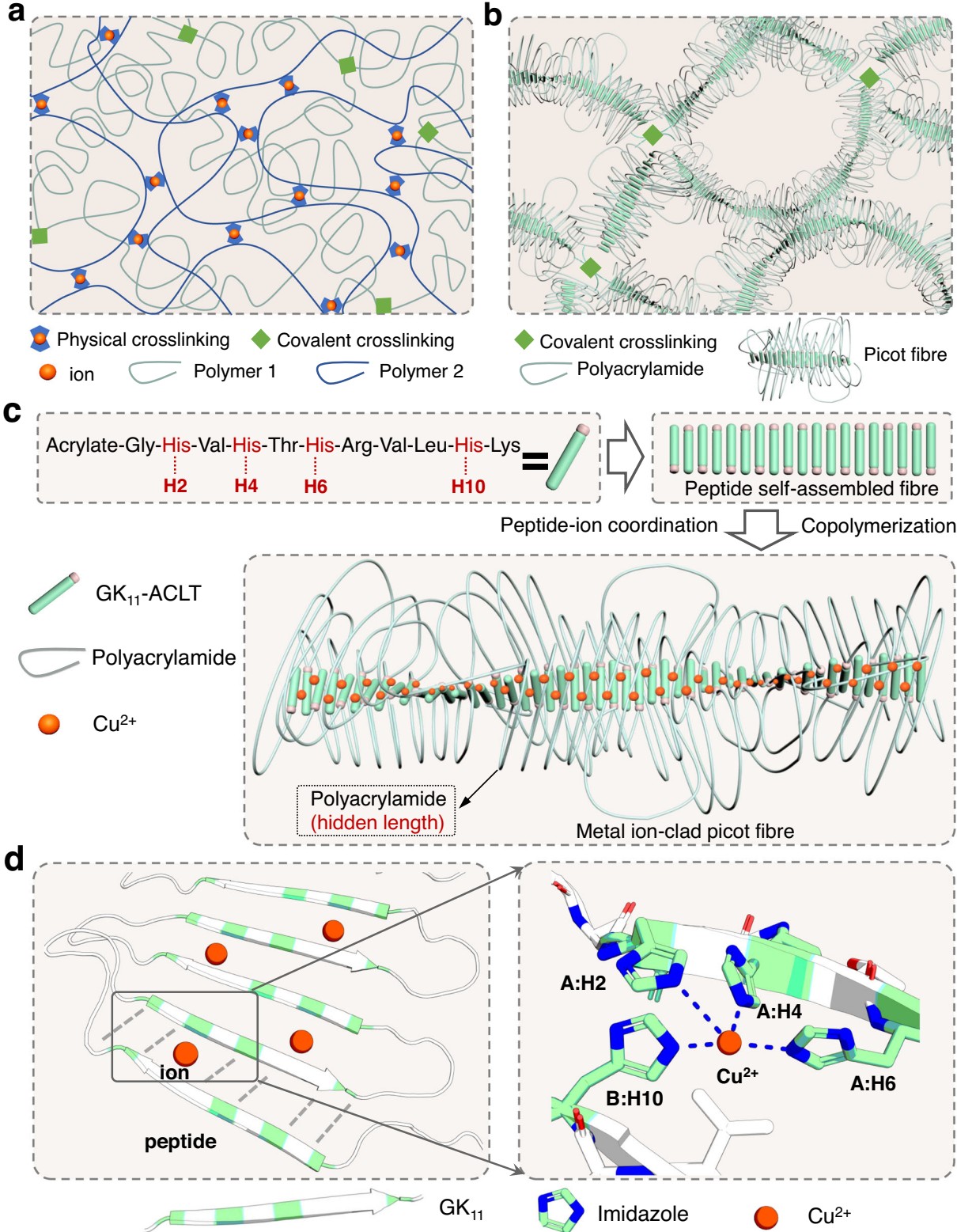

**Fig. 1 | Hydrogel design based on metal ion-clad picot fibres. a** Schematic of a conventional double-network hydrogel. The physically crosslinked network is designed to dissipate energy, and the covalently crosslinked network is used to provide hidden length. **b** Schematic of the hydrogel constructed by picot fibres made of self-assembling peptide strands with zipped flexible hidden lengths. Upon deformation of the hydrogel, the picot fibres are extended to dissipate energy efficiently and release the polyacrylamide hidden length without reducing the network connectivity of the hydrogels. **c** The peptide sequence and synthetic scheme for the metal ion-clad picot fibres. H2, H4, H6 and H10 correspond to four histidine residues at different positions of a $GK_{11}$ peptide. **d** The self-assembled structure of $GK_{11}$ (left) and a magnified region of this diagram showing the binding position of $Cu^{2+}$ (right). The hydrogen bonds are shown as grey dashed lines. Histidine is coloured green in the ribbon representation in the left image and represented by sticks to illustrate the coordination bonds in the right image. A: H2, A: H4 and A: H6 correspond to three histidine residues in the N-terminal of a peptide and B: H10 corresponds to the histidine in the C-terminal of the neighbouring peptide.

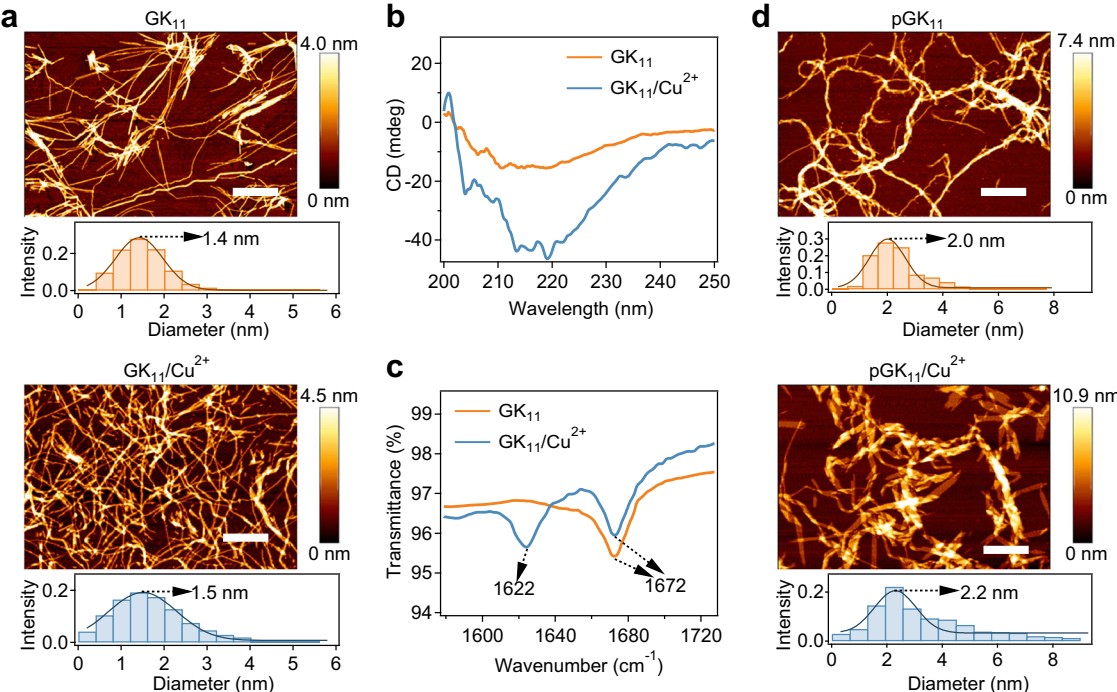

**Fig. 2 | Characterisation of supramolecular and picot GK₁₁ peptide fibres without and with metal Cu²⁺ coordination. a** AFM images of the fibres formed by the self-assembly of GK₁₁ in the absence (top) and presence (bottom) of Cu²⁺. The graphs below the images correspond to the diameter distributions of the fibres.

Scale bar = 1 μm. **b**, **c** CD spectrum (**b**) and FT-IR spectroscopy (**c**) of the self-assembled GK₁₁ peptide with (GK₁₁/Cu²⁺) and without Cu²⁺ (GK₁₁). **d** As for (**a**), but for picot fibres (top) and metal ion-clad picot fibres (bottom) formed by GK₁₁ linked with polyacrylamide in the absence and presence of Cu²⁺. Scale bar = 1 μm.

further revealing that most of the acrylamide was integrated into the hidden length of picot fibres.

## Hydrogels made of metal ion-clad picot fibres

Having successfully prepared pGK₁₁/Cu²⁺ fibres, we next used them as building blocks to construct hydrogels. The hydrogels were prepared in a two-step process. First, the precursor hydrogels were synthesised via a one-pot copolymerisation of acrylamide, acrylate-terminated four-armed polyethylene glycol (four-armed PEG-ACLT) and ACLT-GK₁₁ peptide in deionized water. Here, four-armed PEG-ACLT was used as the covalent crosslink of the network (Fig. 1b). More than 98% of the peptide was successfully integrated into the hydrogels according to the UV-vis spectrum (Supplementary Fig. 7). Then, the hydrogels were dialysed in Tris buffer containing Cu²⁺ ions to remove unreacted monomers and bind the fibres with Cu²⁺. The obtained hydrogels made of pGK₁₁/Cu²⁺ fibres are denoted p-Pep/Cu²⁺ hydrogels. The light blue colour of the p-Pep/Cu²⁺ hydrogels indicated the formation of GK₁₁-Cu²⁺ coordination complexes (Fig. 3a). Hydrogels containing only PEG and acrylamide (denoted PAM hydrogel) and hydrogels containing PEG, acrylamide, and pGK₁₁ fibres (denoted p-Pep hydrogel) were also prepared as controls. Self-assembly of the GK₁₁ inside the hydrogels was confirmed by the similar FT-IR peaks of the hydrogels and the peptide fibres (Supplementary Fig. 8). The swelling ratio of the p-Pep hydrogel (343.6%) was smaller than that of the PAM hydrogel (409.6%) due to formation of the peptide fibres (Supplementary Fig. 9a). Meanwhile, the swelling ratio of the p-Pep/Cu²⁺ hydrogel (209.2%) was even smaller, due to the effects of GK₁₁-Cu²⁺ coordination. The water contents of all the hydrogels were higher than 87% (Supplementary Fig. 9b).

Due to the formation of metal ion-clad picot fibres, the p-Pep/Cu²⁺ hydrogel was more compressible than the PAM and p-Pep hydrogels. As shown in Fig. 3a and Supplementary Movie 1, the compression limits for the p-Pep and PAM hydrogels were less than 90% and 80%, respectively, whereas the p-Pep/Cu²⁺ hydrogel almost fully recovered

from a strain of 90%. The p-Pep/Cu²⁺ hydrogel can even recover without any permanent damage after being compressed with a sharp blade (Fig. 3b and Supplementary Movie 2). In addition to its excellent compressive properties, the p-Pep/Cu²⁺ hydrogel slide was stretched to more than 23 times its initial length (Fig. 3c). The deformability of p-Pep/Cu²⁺ hydrogels can be mainly attributed to the zipped acrylamide in the hidden length of the peptide fibres, which also contributed to the low swelling ratio. The released hidden length of peptide fibres during deformations led to significantly enhanced deformability. Moreover, the p-Pep/Cu²⁺ hydrogel can recover instantly even after 1000 cycles of stretching (strain -15 mm mm⁻¹) at a frequency of 0.5 Hz without showing any obvious mechanical fatigue (Supplementary Fig. 10 and Supplementary Movie 3).

## Strength, stiffness and toughness

Subsequently, the mechanical properties of the hydrogels were quantitatively determined with standard tensile mechanical tests. Typical tensile stress–strain curves of PAM, p-Pep, and p-Pep/Cu²⁺ hydrogels are shown in Fig. 4a, and the details of the mechanical properties are summarised in Supplementary Table 1. Unless otherwise stated, the peptide concentrations in the p-Pep and p-Pep/Cu²⁺ hydrogels were 6% w/v. The fracture stress, Young's modulus and the work of rupture ($W_r$) for the p-Pep/Cu²⁺ hydrogel were significantly higher than those of the PAM and p-Pep hydrogels (Fig. 4a and Supplementary Table 1). The Young's modulus of the PAM, p-Pep and p-Pep/Cu²⁺ hydrogels were 13.2, 25.4 and 71.5 kPa, respectively. Since all of the hydrogels contained the same concentration of covalent crosslinks (four-armed PEG-ACLT), the significantly increased Young's modulus values for the p-Pep and p-Pep/Cu²⁺ hydrogels were presumably due to the presence of the picot fibres. The stress–strain curves did not show any yielding points that are typically observed in double-network hydrogels because elements of the hidden length were supposed to release one at a time upon rupture of neighbouring β strands in the picot

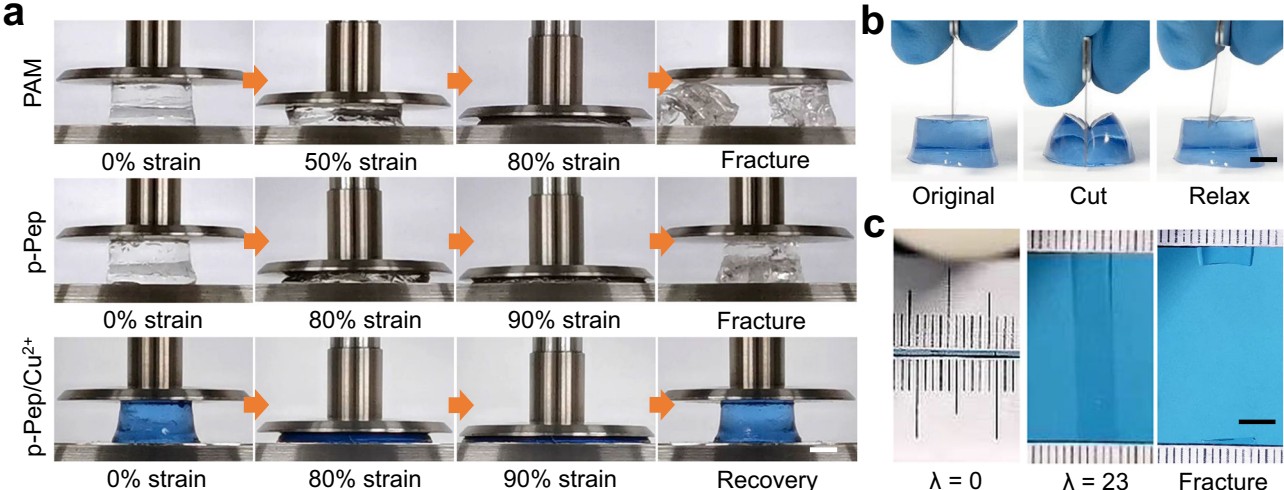

**Fig. 3 | Hydrogels made of metal ion-clad picot fibres. a** Photographs of the PAM (top), p-Pep (middle) and p-Pep/Cu$^{2+}$ (bottom) hydrogels under compression-relaxation cycles. The PAM and p-Pep hydrogels fractured at strains of 80 and 90%, whereas the p-Pep/Cu$^{2+}$ hydrogel almost fully recovered from a strain of 90%. Scale bar = 5 mm. **b** Photographs of a p-Pep/Cu$^{2+}$ hydrogel compressed with a sharp blade and allowed to relax. No detectable cut was observed on the gel after the release of compression. Scale bar = 5 mm. **c** Photographs of the original, stretched (strain = 23 mm mm$^{-1}$) and fractured p-Pep/Cu$^{2+}$ hydrogel. The original width and length of the sample between the clamps were 8 and 1 mm, respectively. Scale bar = 5 mm. For **a**–**c**, the peptide concentration in the precursors of the p-Pep and p-Pep/Cu$^{2+}$ hydrogels was 6% w/v.

fibres (Supplementary Fig. 11); this behaviour was distinct from strain-dependent stretching of the networks in double-network hydrogels[48].

The work of rupture of PAM, p-Pep and p-Pep/Cu$^{2+}$ hydrogels were ~0.4, 7.3 and 38.1 MJ m$^{-3}$, respectively. The Young's modulus increased with increasing GK$_{11}$ concentrations in the p-Pep/Cu$^{2+}$ hydrogels (Fig. 4b). The fracture stress and work of rupture also increased at increasing GK$_{11}$ concentrations at low peptide concentrations up to 6% and dropped when further increasing the peptide, presumably due to the aggregation of the peptide fibres (Supplementary Fig. 12). Moreover, the toughness (fracture energy)[29,49] was 25.3 kJ m$^{-2}$ for the p-Pep/Cu$^{2+}$ hydrogels, much higher than the 2.6 kJ m$^{-2}$ for the p-Pep and 0.1 kJ m$^{-2}$ for the PAM hydrogels (Supplementary Fig. 13) and also comparable to those of many other hydrogels reported in the literature (Supplementary Table 2). For hydrogels with normal network structures, stiffness is linearly proportional to the crosslinking density[50], and toughness is inversely proportional to the square root of the crosslinking density[51,52]. Therefore, both chemical and physical crosslinks stiffen hydrogels with the trade-off of reducing their toughness. In contrast, in the p-Pep/Cu$^{2+}$ hydrogels, the physical bonds were mainly used to build the metal ion-clad picot fibres. Each fibre can be considered an extensible crosslinker. Increasing physical interactions by raising the peptide or metal ion concentrations only increases the total available sacrificial bonds in the fibres without affecting the crosslinking density of the hydrogels, thus resolving the stiffness-toughness trade-off common in polymer networks. Note that Young's modulus and fracture strain of the p-Pep/Cu$^{2+}$ hydrogel varied with the deformation rates (Supplementary Fig. 14), which indicates that rupture of the metal ion-clad picot fibres in the hydrogels was a none-quilibrium process[8,53].

Having established the unique toughening mechanism of our hydrogels, we then systematically optimised the mechanical properties by adjusting the concentrations of 4-armed PEG and acrylamide in the hydrogels (Supplementary Fig. 15). The fracture strain and work of rupture of the p-Pep/Cu$^{2+}$ hydrogels reached more than 2500% and 50 MJ m$^{-3}$ by simply increasing the concentration of acrylamide (Supplementary Fig. 15c, d). Work of rupture and strength of hydrogels were also enhanced by the picot fibres in the compressive mechanical tests (Supplementary Fig. 16). Notably, the maximum compression

strain and maximum load of p-Pep/Cu$^{2+}$ hydrogels reached more than 95% and 1.4 MPa, revealing the remarkable compressibility. All of these results demonstrate the excellent mechanical performance enabled by incorporating the metal ion-clad picot fibres into the hydrogel networks.

To further confirm the synergistic contributions of peptide self-assembly and metal coordination to the mechanical properties of the p-Pep/Cu$^{2+}$ hydrogels, we investigated how the mechanical properties of the hydrogels were affected by perturbing the structures of the picot fibres. We used ethylenediamine tetraacetic acid (EDTA) to remove Cu$^{2+}$ ions, which led to considerable decreases in the values of Young's moduli and dissipated energy (Supplementary Fig. 17a, b). The addition of a denaturant, guanidium chloride (GuHCl), fully unfolded the picot fibres and destroyed the Cu$^{2+}$ binding sites, which led to further deterioration of the mechanical properties (Supplementary Fig. 17a, b). However, the mechanical properties of the hydrogels were recovered by removing the EDTA or denaturant and recharging the picot fibres with Cu$^{2+}$ (Supplementary Fig. 17c–f). At low strains, the peptide fibres remain integrated without unfolding. Thus, the poly-acrylamide remains at the hidden length and does not contribute to the elasticity of the p-Pep/Cu$^{2+}$ hydrogels. The hydrogels are formed by four-armed PEG-ACLT with the picot fibres as unextendible cross-linkers. The elasticity of the hydrogels can be predicted using the classical rubber elasticity theory. According to the classical rubber elasticity theory and Gaussian chain statistics[54–57], the molecular weight of a segment between two neighbouring crosslinking points is proportional to $\rho/E$, where $\rho$ is the density of the polymer in the network and $E$ is Young's modulus. The significant change in Young's modulus before and after the addition of GuHCl indicated that the hidden length released between two neighbouring crosslinks in the hydrogels was $1.8 \pm 0.3$ times the initial length, which was close to the theoretical value of 2.0 (Supplementary Fig. 18). Therefore, the mechanical properties of the hydrogels were consistent with the proposed structure. It is worth noting that at high strains, the elasticity of the hydrogels can be affected by the rupture of the peptide fibres as well as the alignment of the fibres to the force direction, which is quite complicated. Moreover, the unfolding of the peptide fibres contributes largely to the fracture strain and toughness of the hydrogels.

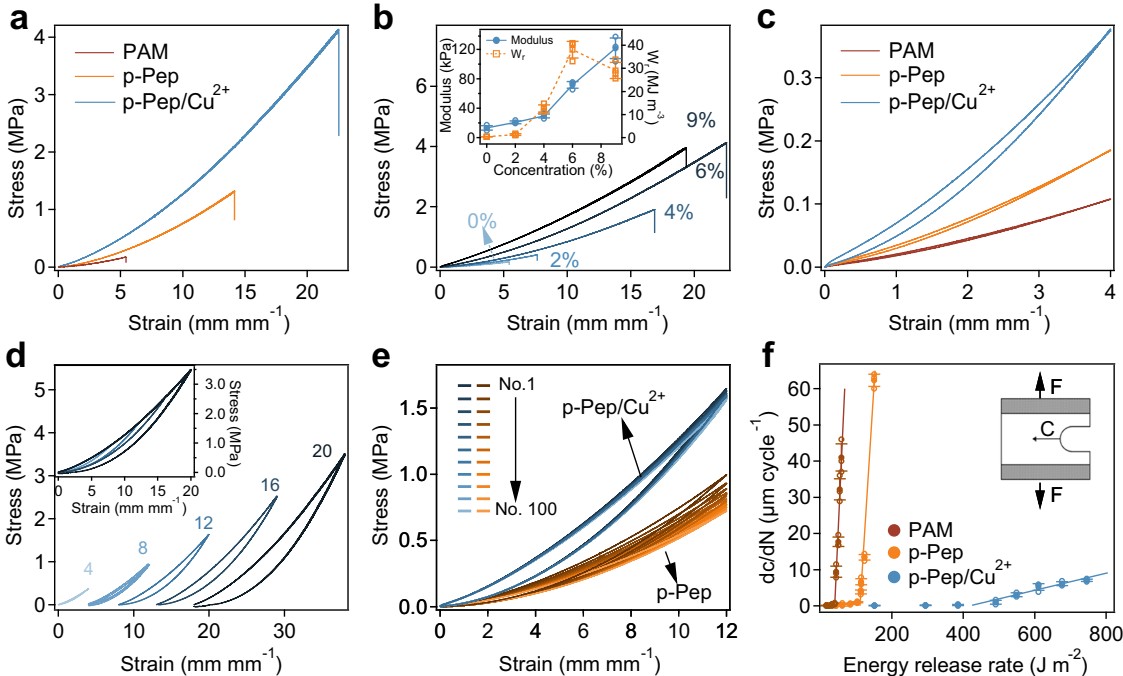

**Fig. 4 | Mechanical properties of hydrogels made of metal ion-clad picot fibres.** **a** Typical stress–strain curves under tension for PAM, p-Pep and p-Pep/Cu²⁺ hydrogels. **b** Typical strain–stress curves of p-Pep/Cu²⁺ hydrogels under tension at various peptide concentrations in the precursor of hydrogels (0%, 2%, 4%, 6% and 9% w/v). The inset corresponds to the summarised Young's modulus and work of rupture ($W_r$) of the p-Pep/Cu²⁺ hydrogels. Values represent the mean and standard deviation ($n = 3$ independent samples). **c** Typical stretching-relaxation cycles of PAM, p-Pep and p-Pep/Cu²⁺ hydrogels at a strain of 4 mm mm⁻¹. **d** Stretching-relaxation cycles for p-Pep/Cu²⁺ hydrogels subjected to various strains (4, 8, 12, 16 and 20 mm mm⁻¹). The curves are offset for clarity, and the overlapping curves are

shown in the inset image. **e** Stress–strain curves for 100 consecutive stretching-relaxation cycles without any stops for p-Pep and p-Pep/Cu²⁺ hydrogels. The cycle numbers are 1, 10, 20, 30, 40, 50, 60, 70, 80, 90 and 100. **f** Extension of cracks per cycle as a function of the energy release rate for the PAM, p-Pep and p-Pep/Cu²⁺ hydrogels. The inset illustrates the stretching of notched samples. The $R^2$ values of the linear fitting to estimate fatigue thresholds for PAM, p-Pep and p-Pep/Cu²⁺ hydrogels were 0.99, 0.99 and 0.95, respectively. The data points represent the mean and standard deviation ($n = 3$ independent experiments). For **a**, **c**, **d**–**f**, the peptide concentration in the hydrogel precursors was 6% w/v.

## Hysteresis and recovery

When hydrogels are subjected to cyclic loading/unloading at speeds faster than the spontaneous dissociation/association dynamics of the physical crosslinks, the stress–strain curves show hysteresis due to energy dissipation[58]. The tensile stress–strain curves for the p-Pep and p-Pep/Cu²⁺ hydrogels showed clear hysteresis, while little hysteresis was observed for PAM hydrogels (Fig. 4c and Supplementary Fig. 19a). Increasing the metal ion-clad picot fibre contents in the hydrogels led to increased energy dissipation (Supplementary Fig. 19b, c). The same trends were observed for the compressing-relaxation cycles (Supplementary Fig. 20a, b). Moreover, hysteresis for the p-Pep/Cu²⁺ hydrogel was enhanced at larger strain, indicating that more picot fibres were unfolded at larger strains (Fig. 4d and Supplementary Figs. 19d, 20c, d). Together, these experiments confirmed the role of the metal ion-clad picot fibres in the reversible energy dissipation of the hydrogels.

Next, the recovery of the hydrogels was studied by applying 100 sequential stretching/relaxation cycles. The stress–strain curves of the p-Pep/Cu²⁺ hydrogels almost fully overlapped, while the stress of the p-Pep hydrogels decreased gradually with an increasing number of cycles (Fig. 4e). For p-Pep/Cu²⁺ hydrogels, the maximum stress and the total dissipated energy per cycle of the p-Pep/Cu²⁺ hydrogels remained 95% and 92% at a strain of 12 mm mm⁻¹, even after they were continuously stretched for 100 cycles (Supplementary Fig. 21a); this suggests that rapid recovery of the baseline mechanical properties occurred within seconds under dynamic loading. In contrast, the maximum stress and the energy dissipated per cycle of p-Pep decreased to 72 and 60%, respectively, after 100 cycles of stretching. Similar trends were observed in the stress–strain curves for 100 compressing/relaxation cycles (Supplementary Fig. 21b–d). We

attribute the ultrafast recovery of the mechanical properties to localised energy dissipation and recovery. Reformation of the picot fibres involved only the recovery of intramolecular interactions and was a localised process that was independent of recovery for the whole hydrogel network. In contrast, in many hydrogels with physical crosslinks[18,26,28], the ruptured crosslinks are prone to rebinding at incorrect positions due to the slow relaxation of the hydrogel network. The misregistered crosslinks slow the kinetics associated with the full recovery of the structure and mechanical properties of the hydrogels. Moreover, the p-Pep/Cu²⁺ hydrogels recovered even faster than the p-Pep hydrogels. It is probably because Cu²⁺ remained attached to the unfolded picot fibres and thus facilitated the reformation of the picot fibres in the unloaded hydrogels, as indicated by the almost unchanged mechanical strength and negligible Cu²⁺ release of p-Pep/Cu²⁺ hydrogels under stretching-relaxation cycles in Tris buffer (Supplementary Fig. 22). These Cu²⁺ ions may serve as the nuclei for the folding of picot fibres, as the fibres without Cu²⁺ showed reduced recovery rates.

## Fatigue resistance

Fatigue resistance is also associated with the energy dissipation mechanism and reflects the ability of the hydrogel to resist fatigue fracture under repeated mechanical loads. The fatigue resistance properties of the hydrogels were evaluated using the procedures reported by refs. 32,59. Supplementary Fig. 23a–c shows images of the precut PAM, p-Pep and p-Pep/Cu²⁺ hydrogels after stretch-relaxation cycles at a strain of 1.0 mm mm⁻¹. The rate of crack propagation in the p-Pep/Cu²⁺ hydrogels was much slower than those of the PAM and p-Pep hydrogels at the same strain. Specifically, in the PAM hydrogel,

the cracks extended rapidly and propagated throughout the hydrogel within 300 cycles (Supplementary Fig. 23a). By contrast, in the p-Pep hydrogel, the cracks propagated slowly throughout the hydrogel within 3400 cycles (Supplementary Fig. 23b). In sharp contrast, no obvious crack propagation was observed for the p-Pep/Cu²⁺ hydrogel even after 10,000 cycles (Supplementary Fig. 23c). Then, we applied cyclic stretching-relaxation at different strains to precut hydrogels and recorded the crack propagation rates to calculate the fatigue thresholds of the hydrogels (Supplementary Fig. 23d–f). Based on these data, the propagation of cracks per cycle, $dc/dN$, vs. the energy release rate, $G$, are plotted in Fig. 4f, and a linear regression was used to estimate the fatigue threshold. The fatigue thresholds of the PAM and p-Pep hydrogels were ~40 ± 0.1 J m⁻² and ~111 ± 3 J m⁻², respectively. In contrast, the fatigue threshold of the p-Pep/Cu²⁺ hydrogel was ~424 ± 22 J m⁻², which is much higher than those of various tough hydrogels[31,60–62]. A recent study showed that dynamic interactions can only carry load and contribute to fatigue resistance when their lifetimes are longer than the timescale of the periodic mechanical load[30]. Note that both the lifetimes and recovery rates of typical physical interactions are considerably reduced by applied force[63,64]. Therefore, they cannot carry a load under cyclic stretch to prevent crack propagation. We attribute the excellent fatigue resistance of the p-Pep/Cu²⁺ hydrogel to the high mechanical stability and fast reassembly of the metal ion-clad picot fibres. In addition, the release of hidden length can also prevent crack propagation and increase the fatigue threshold according to the Lake-Thomas model[51,65]. This finding was consistent with a recent report that highlighted the importance of embedded hidden length on the fatigue resistance of double-network hydrogels[33].

In this study, we introduced the use of self-assembled picot peptide fibres as building blocks to engineer hydrogels with high strength, toughness, fatigue resistance and rapid mechanical recovery. In this design, the hidden length is embedded in the peptide fibres instead of being provided by an additional uncrosslinked network, which resolves the stiffness-toughness trade-off common in polymer networks. Moreover, the self-assembled peptide fibres clad with metal ions were mechanically stable yet reformed quickly due to cooperative interactions in the self-assembled structures. Therefore, the resulting hydrogel showed a combination of high strength (~4.1 MPa), good toughness (25.3 kJ m⁻²), a high fatigue threshold (~424 J m⁻²) and rapid recovery (seconds). Our study highlights the importance of tailoring hydrogel network structures at the molecular level to achieve uncharted mechanical performance. It is possible to further improve the mechanical properties by using different self-assembling peptides or synthetic motifs. We anticipate that the engineered hydrogels demonstrated in this study may find broad applications as tissue engineering scaffolds, stretchable sensors, and components of soft robotics.

## Methods

### Materials
Acrylamide (purity: 99%), ammonium persulfate (APS, purity: 98%), ethylenediamine tetraacetic acid (EDTA, purity: 99%), guanidine hydrochloride (GuHCl, purity: 99%), sodium diethyldithiocarbamate (DDTC-Na, purity: 98%) and tris(hydroxymethyl)methyl amino-methane (Tris, purity: 99%) were purchased from Sigma Aldrich Inc. Glycine-histidine-valine-histidine-threonine-histidine-valine-leucine-histidine-lysine (GK₁₁, purity: 95%) and acrylate-histidine-valine-histidine-threonine-histidine-arginine-valine-leucine-histidine-lysine (ACLT-GK₁₁, purity: 95%) were purchased from GL Biochem (Shanghai, China). Acrylate-terminated four-armed polyethylene glycol (four-armed PEG-ACLT, 20 kDa, purity: 99%) was purchased from SINOPEG (Xiamen, China). Unless otherwise stated, all the other reagents were purchased from Aladdin Inc. All materials were used without further purification.

### Atomic force microscope (AFM) imaging and nanoindentation
The AFM samples were prepared as follows: For the preparation of GK₁₁ and GK₁₁/Cu²⁺ nanofibres, the GK₁₁ peptide was dissolved in 1 M Tris buffer (pH = 7.60, containing 300 mM KCl) with and without CuCl₂ (45 mM) to a concentration of 45 mM. The solutions were stored at room temperature (22 °C) for 2 h to allow the self-assembly of peptides and coordination between the peptide and Cu²⁺. Then, the dispersion was diluted to a peptide concentration of 0.5 mg mL⁻¹. About 50 μL of the freshly prepared solution was loaded onto a freshly peeled mica surface and allowed to adsorb for 5 min. After removing the majority of the solutions, the samples were dried at room temperature (22 °C). For the preparation of picot fibres, ACLT-GK₁₁ (43.2 mM) and acrylamide (703 mM or 1406 mM) were copolymerised in deionized water and dissolved in 1 M Tris buffer (pH = 7.60, containing 300 mM KCl) to induce self-assembly. For the preparation of metal ion-clad picot fibres, the polymer was diluted three times using the Tris buffer containing 150 mM CuCl₂ to induce self-assembly and ion binding. Then 50 μL of the solution was loaded onto a freshly peeled mica surface and allowed to adsorb for 5 min. After removing the majority of the solutions, the samples were dried at room temperature (22 °C).

Finally, the samples were imaged using a NanoWazrid IV (JPK. Germany) in a QI mode (conditions: pixels: 512 × 512; Z length: 0.1 μm; extend and retract speed: 30 μm s⁻¹; Z resolution: 80,000 Hz; maximum loading force: 130 nN). Silica cantilevers (PPP-SeIHR, NanoSensors, half-open angle of the pyramidal face of θ: <10°, tip radius: ~10 nm, spring constant: ~42 N m⁻¹) were used in all experiments. The point stiffness was determined as the normal force divided by the deformation of the sample and calculated from the force-displacement curves. For each sample, more than six regions were randomly selected to perform the nanoindentation. At least three cantilevers were used in the experiments to exclude a tip dependency on the results. All the data were analysed, and the two-dimensional diagrams were reconstructed using the JPK data processing 7.0.46 software (JPK company).

### Preparation of hydrogels
In a typical preparation of the p-Pep/Cu²⁺ hydrogels, acrylamide and 4-armed PEG-ACLT (MW: 20 KDa) were dissolved in Milli-Q water at 225 mg mL⁻¹ (3165.5 mM) and 50 mg mL⁻¹ (2.5 mM). Then, the ACLT-GK₁₁ peptide was dissolved in the mixture to the concentration of 60 mg mL⁻¹ (43.2 mM). The precursor solution was degassed with argon for 3 × 15 min to remove dissolved oxygen. Then, the solutions were stored at room temperature (22 °C) for 2 h to allow the self-assembly of peptides. Ammonium persulfate (APS) was added as the photoinitiator, and the polymerisation was conducted under UV (285 nm) illumination at room temperature (22 °C) for 20 h. Transparent pregels were obtained and extensively washed with 1 M Tris buffer (pH = 7.60, containing 300 mM KCl) for 24 h to remove the unreacted monomers. Then, the hydrogels were dialysed in 1 M Tris buffer (pH = 7.60, containing 300 mM KCl and 200 mM CuCl₂) for 24 h to trigger the formation of ion binding. Then, the hydrogels were dialysed in 1 M Tris buffer (pH = 7.60, containing 300 mM KCl) for 24 h to remove the redundant CuCl₂ and stored at 4 °C before measurements.

For the preparation of p-Pep hydrogel, acrylamide and four-armed PEG-ACLT (MW: 20 KDa) were dissolved in Milli-Q water at 225 mg mL⁻¹ (3165.5 mM) and 50 mg mL⁻¹ (2.5 mM). Then, the ACLT-GK₁₁ peptide was dissolved in the mixture to the concentration of 60 mg mL⁻¹ (43.2 mM). The precursor solution was degassed with argon for 3 × 15 min to remove dissolved oxygen. Then, the solutions were stored at room temperature (22 °C) for 2 h to allow the self-assembly of peptides. Ammonium persulfate (APS) was added as the photoinitiator, and the polymerisation was conducted under UV (285 nm) illumination at room temperature (22 °C) for 20 h. Transparent pregels were obtained and dialysed in 1 M Tris buffer (pH = 7.60, containing 300 mM KCl) for 24 h to remove the unreacted monomers and achieve the swelling equilibrium.

For the preparation of PAM hydrogel, acrylamide and four-armed PEG-ACLT (MW: 20 KDa) were dissolved in Milli-Q water at 225 mg mL$^{-1}$ (3165.5 mM) and 50 mg mL$^{-1}$ (2.5 mM). The precursor solution was degassed with argon for $3 \times 15$ min to remove dissolved oxygen. Ammonium persulfate (APS) was added as the photoinitiator, and the polymerisation was conducted under UV (285 nm) illumination at room temperature (22 °C) for 20 h. Transparent pregels were obtained and dialysed in 1 M Tris buffer (pH = 7.60, containing 300 mM KCl) for 24 h. Unless otherwise stated, the ACLT-GK$_{11}$ peptide concentration of 60 mg mL$^{-1}$ (6% w/v) was used in the preparation of p-Pep and p-Pep/Cu$^{2+}$ hydrogels.

## Tensile and compressive test

The tensile/compressive stress–strain measurements were performed using a tensile-compressive tester (Instron-5944 with a 2 kN sensor) in the air at room temperature (22 °C). A humidifier was applied to keep the hydrogel wet during all the measurements. In the tension-crack and stretching-relaxation tests, the strain rate of stretching was maintained at ~0.5–1.0 mm mim$^{-1}$. In the compression-crack and compressing-relaxation tests, the rates of compression were maintained at 0.8–1.0 mm min$^{-1}$. The work of rupture ($W_r$) was calculated from the area below the tensile stress–strain curve until fracture. In addition, Young's modulus for tension and compression comprised the approximate linear fitting values in the strain of 0–1, and 0–0.2 mm mm$^{-1}$, respectively. For the recovery experiments of the hydrogels, cyclic loading/unloading (stretching rate 10 mm min$^{-1}$) was applied to the same hydrogel and the stress–strain curves were recorded[20,26,66].

For the measurement of energy release rates, a hydrogel with a crack being created to the edge was subjected to continuous stretching-relaxation cycles at the strain of $\lambda$. The crack propagation was recorded using a camera from the undeformed hydrogels before and after the Nth cycle. Then the crack propagation was divided by the number of cycles to get the extension of crack per cycle, dc/dN, assuming that c linearly increases in N cycles. The loading rate was ~0.5 Hz. The width and length of the hydrogel between the grippers were ~12 and 3.5 mm, respectively. The initial length of the crack was 2.4–3.0 mm. The energy release rate $G$ of the notched hydrogel is determined as $G = H(\lambda)$, where $H$ is the distance between the two grippers of the tensile tester when the notched hydrogel is undeformed, $(\lambda)$ is the energy per volume of the unnotched sample while stretched.

## Data availability

All data were available in the main text, Supplementary Information and Supplementary Movies. All data were available upon request. Source data are provided with this paper.

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

## Acknowledgements

This work is supported mainly by the National Key R&D Programme of China (Grant No. 2020YFA0908100, Y.C.), the National Natural Science Foundation of China (No. 11934008, W.W., T2225016, Y.C., 12002149, Y.L. and 11974174, M.Q.) and the Natural Science Foundation of Jiangsu Province (No. BK20220120, B.X., BK20221437, Y.G.).

## Author contributions

Y.C. and W.W. conceived the idea and designed the study. B.X. and Z.B. performed the experiments and analyzed the results. W.Y., W.S. and Y.L. prepared some of the hydrogel samples. Y.G. and Y.Z. performed the simulations. Y.C., W.W. and B.X. wrote and refined the paper. Y.C., W.W. and M.Q. supervised the project. All the authors discussed the results.

## Competing interests

The authors declare no competing interests.
