## [Peer Review File · Nature Communications]

Strong, tough, rapid-recovery, and fatigue-resistant hydrogels made of picot peptide fibresREVIEWER COMMENTS

Reviewer #1 (Remarks to the Author):

Xue et al. have developed a novel hydrogel based of a peptide-polymer composite with coordinating metal ions. They created a hybrid bio-material with toughness and resistance to fracture. To increase the strength of the hydrogel, Cu^{2+} ions were added and coordinated between histidine residues between antiparallel peptide strands, resulting in a beta-sheet structure. The hydrogels were subjected to stress, fatigue, and hysteresis experiments.

Specific Comments:

Hidden length was not discussed in the introduction. A brief sentence or two explaining why hidden length is important for increasing toughness would be helpful.

In figure 1D, can you offer any rationale for why a second Cu^{2+} is not coordinated between 2 adjacent strands? Also, consider inverting the color scheme (i.e. green for histidines and white for other residues).

At the end of section “Design of hydrogels based on metal ion-clad picot fibres”, the claim is made that the binding energy between the GK11 peptide and Cu^{2+} was -7.2 kcal/mol and is five times the strength of a typical hydrogen bond in proteins. How is this measurement determined? Is it calculated computationally or is this experimentally determined. Moreover, is the ‘five times’ comparison being made a one-to-one comparison by experiment/simulation? The authors should add relevant citation(s) at the end of this claim.

The limit of fracture stiffness is presumably due to aggregation of the peptide fibers. I think this is likely, but would a TEM show at low GK11 and high GK11 –concentrations confirm this?

Please briefly describe how crack propagation was measured in addition to citing the work by Suo et al.

Can a standard deviation be applied to the “1.8 times the initial length” value given for the change in Young’s modulus before and after to make a more convincing closeness to the theoretical 2.0 value?

Is there any way to clarify if the faster recovery of Cu^{2+} bound p-Pep hydrogels is due to Cu^{2+} attached remnants?

R^2 values need to be provided for the linearly regressed fatigue threshold limits. The figures make it very difficult to understand the goodness of fit. Also are these measurements in replicates? There is no standard deviation in the plots. Overall, it is difficult to assess confidence in these extrapolations and comparisons to various tough hydrogels.

Reviewer #2 (Remarks to the Author):

The authors report the p-Pep/Cu²⁺ hydrogel having polyacrylamide skeletal network with polypeptide motifs. The polypeptides form supramolecular peptide fibers inside the gel, which can be further reinforced by Cu²⁺ ions. The obtained gels exhibit large stretchability, high toughness, and fast recovery. This material belongs to tough soft materials based on the sacrificial bond principle. While typical materials based on this principle have two components, which are the brittle sacrificial component and stretchy matrix, the authors try to integrate these two roles into a single network as illustrated in Fig. 1. This idea is novel and interesting.

On the other hand, the reviewer does not think that the actual gel has the structure drawn in Fig. 1. Also, the reviewer is not sure how this work is essentially novel. The p-Pep/Cu²⁺ hydrogel can be classified as dual cross-linking gels with physical and chemical bonds. It has been generally reported that addition of physical bonding to a chemical gel improves stretchability and toughness (ex. K. Mayumi et al., *Ext. Mech. Lett.* 2016, 6, 52). The difference between the authors' gel and the reported dual cross-linking gels is not explained enough.

Overall, the novelty of the ideas and the mechanical properties make this manuscript worthy of publication in *Nature Communications*. On the other hand, the structure and mechanism of good mechanical properties should be reconsidered. The reviewer suggests major revision.

Comments:

1. The illustrations in fig. 1b and fig. S10 are unrealistic and should be extensively modified. While release of the PAAm hidden length after dissociation of the p-Pep/Cu²⁺ fibers possibly occurs in the gel, the gel should not have such ideal structure where PAM "loops" connect the adjacent peptides. According to the procedure, AM and polymerizable peptides were randomly co-polymerized to form the gel.

Regarding to this comment, the term "PAM loops" is not suitable. Instead of "loops", "hidden length" can be used.

2. According to the proposed structure, mechanical properties of the p-Pep/Cu²⁺ hydrogel should be dominated by enthalpic contribution of inter-peptide hydrogen bonding, not by entropic contribution. Thus, analysis of the elasticity based on the classical rubber elasticity is not suitable.

3. As mentioned before, basically the p-Pep/Cu²⁺ hydrogel is a kind of dual cross-linking gels with physical and chemical bonds. In comparison with other gels having physical bonding, what is the key physical factor to obtain the good mechanical properties including fast recovery? The reviewer requests the authors to add more comments about this.

4. For fig. S16, it is not clear how the authors estimate the theoretical value of 2.0. Please show the estimation process. Is it calculated based on the swelling ratio difference?

5. "As these sacrificial bonds cannot reform efficiently, the hydrogels lack a mechanism to prevent crack propagation and are susceptible to fatigue under cyclic loading"; the reviewer is unsure of the relationship between them. Is the slow reformation of sacrificial bonds really the reason of the poor fatigue resistance?

6. "it remains challenging to achieve high strength, high toughness, robust fatigue resistance, and rapid recovery in the same synthetic hydrogel"; please check the two recent papers published in Science, Liu et al., Science 2021, 372, 1078 / Kim et al., Science 2021, 374, 212. These works have realized high toughness of elastic hydrogels (with rapid recovery) with different strategies.

7. The p-Pep/Cu²⁺ hydrogel is more deformable than the PAM gel. While the authors assume that this is due to formation of the peptide fibers, it is also possible that lower swelling ratio of the p-Pep/Cu²⁺ gel ensures its large deformability because swelling of a gel generally decreases its deformability.

8. About fig. S8, use of SEM to estimate mesh size of a wet gel is generally not suitable. Such a "mesh" in a dried gel is probably an artifact and formed during a drying process.

9. Please define the term "toughness" carefully. The authors used the term "toughness" for two different physical quantities. One is the area below the tension stress-strain curve until fracture with the unit of J m⁻³. The other is the fracture energy with the unit of J m⁻², which is called "toughness" in the introduction and conclusion. While the latter should be physically called as toughness, in this field both values are often called as toughness. The reviewer suggests using a different name for the former instead of "toughness" to avoid confusion.

Regarding to this, the authors wrote "toughness is inversely proportional to the square root of the crosslinking density". This "toughness" is the latter one, not the former one.

Point-by-point response to the reviewers' comments

Reviewer #1 (Remarks to the Author):

Xue et al. have developed a novel hydrogel based of a peptide-polymer composite with coordinating metal ions. They created a hybrid bio-material with toughness and resistance to fracture. To increase the strength of the hydrogel, Cu^{2+} ions were added and coordinated between histidine residues between antiparallel peptide strands, resulting in a beta-sheet structure. The hydrogels were subjected to stress, fatigue, and hysteresis experiments.

Specific Comments:

1. Hidden length was not discussed in the introduction. A brief sentence or two explaining why hidden length is important for increasing toughness would be helpful.

Response: We thank the reviewer for the comments. Following the reviewer's suggestion, we have added a brief sentence explaining why hidden length is important for increasing toughness in the introduction. The hidden length in fibril structures or folded proteins can provide extra extensibility and increase the fracture strain after being released, thus greatly enhancing the toughness of hydrogels. The new comments have been included in the revised manuscript. (See the first paragraph on Page 4 in the revised manuscript)

Revisions:

...Moreover, the hidden length in fibril structures or folded proteins can provide extra extensibility after being released, thus enhancing the toughness of hydrogels.³³⁻³⁶...

2. In figure 1D, can you offer any rationale for why a second Cu^{2+} is not coordinated between 2 adjacent strands? Also, consider inverting the color scheme (i.e. green for histidines and white for other residues).

Response: We thank the reviewer for the comments. According to the structure

predicted by AlphaFold2, the β -sheet of a single GK₁₁ peptide is slightly twisted ($\theta \sim 12^\circ$), making side chains of histidine (imidazole) distributing on both sides of the β -sheet in the structure of well-aligned anti-parallel β -sheets. Hence, histidine at the position of H₂, H₄ and H₆ were on the different side of the β -sheet with the histidine at the position of H₁₀ (Fig. 1d). Considering the tetrahedral coordination of Cu²⁺ and histidine, the second Cu²⁺ prefers to form coordination on the other side of the β -sheet with a third strand. Moreover, we also inverted the color scheme, and used green for histidine and white for other residues, following the reviewer's suggestion. (See the first paragraph and Fig. 1 on Page 7 in the revised manuscript)

Revisions:

...Noting that the β -sheet of a single GK₁₁ peptide is slightly twisted ($\theta \sim 12^\circ$), making side chains of histidine (imidazole) distributing on both sides of the β -sheet. Considering the tetrahedral coordination of Cu²⁺ and histidine, the second Cu²⁺ prefers to form coordination on the other side of the β -sheet with a third strand.

Fig. 1. Hydrogel design based on metal ion-clad picot fibres. **a**, Schematic of a conventional double-network hydrogel. The physically crosslinked network is designed to dissipate energy, and the covalently crosslinked network is used to provide hidden length. **b**, Schematic of the hydrogel constructed by picot fibres made of self-assembling peptide strands with zipped flexible **hidden lengths**. Upon deformation of the hydrogel, the picot fibres are extended to dissipate energy efficiently and release the polyacrylamide hidden length without reducing the network connectivity of the hydrogels. **c**, The peptide sequence and synthetic scheme for the metal ion-clad picot fibres. **d**, The self-assembled structure of GK₁₁ (left) and a magnified region of this diagram showing the binding position of Cu²⁺ (right). The hydrogen bonds are shown as grey dashed lines. **Histidine is coloured in green in the ribbon representation in the**

left image and represented by sticks to illustrate the coordination bonds in the right image.

3. At the end of section “Design of hydrogels based on metal ion-clad picot fibres”, the claim is made that the binding energy between the GK₁₁ peptide and Cu²⁺ was -7.2 kcal/mol and is five times the strength of a typical hydrogen bond in proteins. How is this measurement determined? Is it calculated computationally or is this experimentally determined. Moreover, is the ‘five times’ comparison being made a one-to-one comparison by experiment/simulation? The authors should add relevant citation(s) at the end of this claim.

Response: We thank the reviewer for the comments. The binding energy between the GK₁₁ peptide and Cu²⁺ was calculated based on the simulation. The GK₁₁-Cu²⁺ binding structure was first predicted using Alphafold2 neural network through ColabFold pipeline. Then the predicted GK₁₁-Cu²⁺ binding structure was used to calculate the binding free energy with the MM-PBSA method. For all trajectory with Amber15, the decomposition of binding free energy was calculated by considering the enthalpic part containing: Electrostatic, Van der Waals, Polar and Non-Polar Solvation.

$$\Delta G_{bind} = \Delta E_{ele} + \Delta G_{vdW} + \Delta G_{pol} + \Delta G_{nonpol} \quad (1)$$

We have indicated that the binding energy was determined by simulation in the revised manuscript and the simulation details are included in the section of Supplementary Methods in Supplementary Information. (See the first paragraph on Page 2 in the Supplementary Information)

The “five times” comparison is made based on the energy of hydrogen bond previously reported. According the studies of Sheh-Yi et al. and others (*P. Natl. Acad. Sci. USA* 2003, 100, 12683; *P. Natl. Acad. Sci. USA* 1993, 90, 1172), the bond energy of the hydrogen bond in water was ~1.5 kcal/mol, which is one fifth of the binding energy between the GK₁₁ peptide and Cu²⁺ in this work. We have added relevant citations at the end of this claim following the reviewer’s suggestion. (See the last paragraph on Page 6 and references No. 45-46 in the revised manuscript)

Revisions:

...The binding energy between the GK₁₁ peptide and Cu²⁺ was -7.2 kcal mol⁻¹ as determined by simulation, which is about five times the strength of a typical hydrogen bond in proteins^{45,46} ...

4. The limit of fracture stiffness is presumably due to aggregation of the peptide fibers. I think this is likely, but would a TEM show at low GK₁₁ and high GK₁₁ -concentrations confirm this?

Response: We thank the reviewer for the comments. In order to confirm the aggregation of the peptide fibers in hydrogels directly, we investigated the microstructures of p-Pep/Cu²⁺ hydrogels at different GK₁₁ concentrations (2% and 9% w/v) using field emission scanning electron microscope (FESEM) with a resolution of 0.8 nm. It has a similar resolution as TEM but can image hydrogel samples directly without the need of complicated sample slicing. As shown by Supplementary Fig. 12, the fibers aggregated into dense networks with increased entanglements at the high GK₁₁ concentration. The new results have now been included in the revised manuscript and Supplementary Information. (See the last paragraph on Page 14 in the revised manuscript; See the second paragraph on Page 5 and Supplementary Fig. 12 on Page 16 in the Supplementary Information)

Revisions:

...The fracture stress and work of rupture also increased at increasing GK₁₁ concentrations at low peptide concentrations up to 6% and dropped when further increasing the peptide, presumably due to the aggregation of the peptide fibres (Supplementary Fig. 12)...

Field emission scanning electron microscope (FESEM)

The FESEM images were obtained using a Quanta scanning electron microscope (GeminiSEM360, Zeiss, Germany) at 15 kV. The hydrogels were dialyzed in Milli-Q

water for 24 h to remove the unbound salts and lyophilized prior to the measurement.

Supplementary Figure 12. FESEM images of p-Pep/Cu²⁺ hydrogels at different GK₁₁ concentrations (2% and 9% w/v). a, FESEM images of p-Pep/Cu²⁺ hydrogels at the GK₁₁ concentration of 2% w/v. b, FESEM images of p-Pep/Cu²⁺ hydrogels at the GK₁₁ concentration of 9% w/v.

5. Please briefly describe how crack propagation was measured in addition to citing the work by Suo et al.

Response: We thank the reviewer for the comments. Generally, a hydrogel with a crack being created to the edge (20% of the width) was subjected to continuous stretching-relaxation cycles at the strain of λ . The hydrogel crack propagation was recorded using a camera from the undeformed hydrogels before and after the Nth cycle. Then the crack propagation was divided by the number of cycles to get the extension of crack per cycle, dc/dN , assuming that c linearly increases in N cycles. The details of the crack propagation determination were included in “Tensile and compressive test” of the Methods section in the manuscript. (See the last paragraph on Page 25 in the revised manuscript)

Revisions:

For the measurement of energy release rates, a hydrogel with a crack being created to

the edge was subjected to continuous stretching-relaxation cycles at the strain of λ . The crack propagation was recorded using a camera from the undeformed hydrogels before and after the Nth cycle. Then the crack propagation was divided by the number of cycles to get the extension of crack per cycle, dc/dN , assuming that c linearly increases in N cycles...

6. Can a standard deviation be applied to the “1.8 times the initial length” value given for the change in Young’s modulus before and after to make a more convincing closeness to the theoretical 2.0 value?

Response: We thank the reviewer for the comments. Considering the standard deviations of Young’s modulus and swelling ratios, the standard deviation of the ratio for the average length between cross-links after and before the release of the hidden length in experiments was 0.3. According to the calculation of the hidden length in the hydrogel network (See the section of “Calculation of the hidden length in the hydrogel network” on Page 7-9 in the Supplementary Information), the ratio of the average length between cross-links after and before the release of hidden length was calculated as $\varepsilon \frac{\rho_2}{\rho_1}$, in which ε is the ratio of Young’s modulus before and after the release of hidden length, ρ_2 and ρ_1 correspond to the densities of the polymer in the network before and after the release of hidden length. Considering the amount of polymer in the network remained consistent before and after the release of hidden length, the polymer density is inversely proportional to the swelling ratio of hydrogels before and after the addition of GuHCl. The standard deviation has now been applied to the “1.8 times the initial length” value in the revised manuscript. (See the end of Page 16 in the revised manuscript)

Revisions:

...The significant change in Young’s modulus before and after the addition of GuHCl indicated that the hidden length released between two neighbouring crosslinks in the hydrogels was 1.8 ± 0.3 times the initial length, which was close to the theoretical value

of 2.0 (Supplementary Fig. 18).

7. Is there any way to clarify if the faster recovery of Cu^{2+} bound p-Pep hydrogels is due to Cu^{2+} attached remnants?

Response: We thank the reviewer for the comments. Now we have performed experiments to show that most Cu^{2+} remained attached to the peptides during cyclic stretching. First, p-Pep/ Cu^{2+} hydrogels were subjected to stretching-relaxation cycles in 1 M Tris buffer (pH=7.60, containing 300 mM KCl) directly (Supplementary Fig. 22a-b). The maximum stress and energy dissipation of the hydrogel remained 93% and 88% in 100 cycles. The Cu^{2+} release after different cycles were detected using a Cu^{2+} sensor, sodium diethyldithiocarbamate (DDTC-Na) (Supplementary Fig. 22c-f). The total percentage of released Cu^{2+} was less than 0.5% in 100 stretching-relaxation cycles, indicating the negligible Cu^{2+} detachment (Supplementary Fig. 22f). Considering that the recovery of p-Pep/ Cu^{2+} hydrogels was much faster than that of the p-Pep hydrogels, these results suggested that the Cu^{2+} attached remnants contributed to the fast recovery of Cu^{2+} bound p-Pep hydrogels. (See the end of Page 19 in the revised manuscript; See the last paragraph on Page 6 and Supplementary Fig. 22 on Page 24 in the revised Supplementary Information)

Revisions:

Moreover, the p-Pep/ Cu^{2+} hydrogels recovered even faster than the p-Pep hydrogels. It is probably because Cu^{2+} remained attached to the unfolded picot fibres and thus facilitated reformation of the picot fibres in the unloaded hydrogels, as indicated by the almost unchanged mechanical strength and negligible Cu^{2+} release of p-Pep/ Cu^{2+} hydrogels under stretching-relaxation cycles in Tris buffer (Supplementary Fig. 22). These Cu^{2+} ions may serve as the nuclei for the folding of picot fibres, as the fibres without Cu^{2+} showed reduced recovery rates.

Cu^{2+} release from hydrogels

For the detecting of Cu^{2+} release from p-Pep/ Cu^{2+} hydrogels, a hydrogel was subjected

to stretching-relaxation cycles. After different cycles, the hydrogel was immersed into 1 M Tris buffer (pH=7.60, containing 300 mM KCl and 0.3 mM sodium diethyldithiocarbamate (DDTC-Na)) immediately for 30 min. The volume of Tris buffer was five times of the p-Pep hydrogel. Then the UV-vis spectra of the leachates were recorded using a V-550 (JASCO Inc., Japan) spectrophotometer. The absorbance at 452 nm was used to monitor the concentration of Cu^{2+} .

Supplementary Figure 22. Mechanical property and Cu^{2+} release of p-Pep/ Cu^{2+} hydrogels under stretching-relaxation cycles in Tris buffer (pH=7.60, containing 300 mM KCl). a, Typical stress–strain curves for p-Pep and p-Pep/ Cu^{2+} hydrogels under stretching-relaxation cycles. The cycle numbers are 1, 10, 20, 40, 60, 80 and 100. b, Summary of the maximum stress and energy dissipation for p-Pep/ Cu^{2+} hydrogels under stretching-relaxation cycles. c, UV-vis spectra for mixture of sodium diethyldithiocarbamate (DDTC-Na) and Cu^{2+} at different concentrations of Cu^{2+} (1-50

μM). The concentration of DDTC-Na was 250 μM . d, Calibration curve of Cu^{2+} concentrations and $\text{OD}_{452\text{nm}}$ (UV-vis absorbance at 452 nm). e, UV-vis spectra of the hydrogel leachates after different cycles in the presence of DDTC-Na. f, Accumulated percentage of released Cu^{2+} from the p-Pep/ Cu^{2+} hydrogels after different cycles. For b, d and f, values represent the mean and standard deviation ($n=3$ independent experiments).

8. R^2 values need to be provided for the linearly regressed fatigue threshold limits. The figures make it very difficult to understand the goodness of fit. Also are these measurements in replicates? There is no standard deviation in the plots. Overall, it is difficult to assess confidence in these extrapolations and comparisons to various tough hydrogels.

Response: We thank the reviewer for the comments. Now we have provided the R^2 values for the linearly regressed fatigue threshold limits in the figure legend of Fig. 4f. The R^2 values of the linear fitting to estimate fatigue thresholds for PAM, p-Pep and p-Pep/ Cu^{2+} hydrogels were 0.99, 0.99 and 0.95, indicating excellent linearity coefficients. Moreover, these measurements were conducted in replicates following the reviewer's suggestion and the standard deviations are shown in the plots. The standard deviations for all the plots were less than 20%. Taking the R^2 values and the standard deviation into consideration, the fatigue threshold of the p-Pep/ Cu^{2+} hydrogels can be reliably reported as $424 \pm 22 \text{ J m}^{-2}$. Therefore, we are confident to conclude that the fatigue threshold for the p-Pep/ Cu^{2+} hydrogels is much higher than most of the other tough hydrogels. The R^2 values of fitting and standard deviation of the data are included in the revised manuscript. (See Fig. 4f and figure legends of Fig. 4f on Page 17-18 in the revised manuscript)

Revisions:

Fig. 4. Mechanical properties of hydrogels made of metal ion-clad picot fibres. **a**, Typical stress–strain curves under tension for PAM, p-Pep and p-Pep/Cu²⁺ hydrogels. **b**, Typical strain–stress curves of p-Pep/Cu²⁺ hydrogels under tension at various peptide concentrations in the precursor of hydrogels (0%, 2%, 4%, 6%, and 9% w/v). The inset corresponds to the summarized Young’s modulus and **work of rupture (W_r)** of the p-Pep/Cu²⁺ hydrogels. Values represent the mean and standard deviation ($n=3$ independent samples). **c**, Typical stretching–relaxation cycles of PAM, p-Pep and p-Pep/Cu²⁺ hydrogels at a strain of 4 mm mm⁻¹. **d**, Stretching–relaxation cycles for p-Pep/Cu²⁺ hydrogels subjected to various strains (4, 8, 12, 16, and 20 mm mm⁻¹). The curves are offset for clarity, and the overlapping curves are shown in the inset image. **e**, Stress–strain curves for 100 consecutive stretching–relaxation cycles without any stops for p-Pep and p-Pep/Cu²⁺ hydrogels. The cycle numbers are 1, 10, 20, 30, 40, 50, 60, 70, 80, 90 and 100. **f**, Extension of cracks per cycle as a function of the energy release rate for the PAM, p-Pep and p-Pep/Cu²⁺ hydrogels. **The R^2 values of the linear fitting to estimate fatigue thresholds for PAM, p-Pep and p-Pep/Cu²⁺ hydrogels were 0.99, 0.99 and 0.95, respectively. The data points represent the mean and standard deviation ($n=3$ independent experiments).** For **a**, **c**, **d**, **e**, and **f**, the peptide concentration in the hydrogel precursors was 6% w/v.

Reviewer #2 (Remarks to the Author):

The authors report the p-Pep/Cu²⁺ hydrogel having polyacrylamide skeletal network with polypeptide motifs. The polypeptides form supramolecular peptide fibers inside the gel, which can be further reinforced by Cu²⁺ ions. The obtained gels exhibit large stretchability, high toughness, and fast recovery.

This material belongs to tough soft materials based on the sacrificial bond principle. While typical materials based on this principle have two components, which are the brittle sacrificial component and stretchy matrix, the authors try to integrate these two roles into a single network as illustrated in Fig. 1. This idea is novel and interesting.

On the other hand, the reviewer does not think that the actual gel has the structure drawn in Fig. 1. Also, the reviewer is not sure how this work is essentially novel. The p-Pep/Cu²⁺ hydrogel can be classified as dual cross-linking gels with physical and chemical bonds. It has been generally reported that addition of physical bonding to a chemical gel improves stretchability and toughness (ex. K. Mayumi et al., *Ext. Mech. Lett.* 2016, 6, 52). The difference between the authors' gel and the reported dual cross-linking gels is not explained enough.

Overall, the novelty of the ideas and the mechanical properties make this manuscript worthy of publication in *Nature Communications*. On the other hand, the structure and mechanism of good mechanical properties should be reconsidered. The reviewer suggests major revision.

Response: We thank the reviewer for his/her comments that “the novelty of the ideas and the mechanical properties make this manuscript worthy of publication in *Nature Communications*”. We also understand that the structure of the hydrogel network should be further clarified to further highlight the novelty. We thank the reviewer for recommending the paper about using physical bonding to improve the stretchability and toughness of chemical gels (K. Mayumi et al., *Ext. Mech. Lett.* 2016, 6, 52) to our attention and we have cited this excellent paper in the revised manuscript (Reference No. 9). We have revised the manuscript according to the reviewers' valuable comments and hope the reviewer will find the revised manuscript is now acceptable in *Nature Communications*.

Comments:

1. The illustrations in fig. 1b and fig. S10 are unrealistic and should be extensively modified. While release of the PAAm hidden length after dissociation of the p-Pep/Cu²⁺ picot fibers possibly occurs in the gel, the gel should not have such ideal structure where PAM “loops” connect the adjacent peptides. According to the procedure, AM and polymerizable peptides were randomly co-polymerized to form the gel.

Regarding to this comment, the term “PAM loops” is not suitable. Instead of “loops”, “hidden length” can be used.

Response: We thank the reviewer for the comments. We agree with the reviewer that the peptide fibres should not have such ideal structure with PAM “loops” connecting the adjacent peptides. According the preparation procedure of p-Pep/Cu²⁺ hydrogels, the peptide self-assembled into fibres and then polymerized through the copolymerization with acrylamide. As shown the new version of Fig. 1b and Supplementary Fig. 10 (now has been changed into Supplementary Fig. 11), the picots formed by PAM are at different lengths and the peptides connected with each other randomly instead of only adjacent peptides. Besides, we used “hidden length” instead of “loop” in the revised manuscript, following the reviewer’s comments. The new comments as well as revised figures are included in the revised manuscript and Supplementary Information. (See the first paragraph on Page 6 and Fig. 1 on Page 7 in the revised manuscript; See Supplementary Fig. 11 on Page 15 in the revised Supplementary Information; See the changes of “loop” in the abstract on Page 2 and the second paragraph on page 5 in the revised manuscript)

On the other hand, we further confirmed that the acrylamide would zip into the hidden length of the picot peptide fibres using AFM imaging and AFM-based single-molecule force spectroscopy (SMFS). The metal ion-clad picot fibres formed by the copolymerization of GK₁₁ and acrylamide at different ratios of acrylamide and peptide were prepared and scanned using AFM. The diameters of the metal ion-clad picot fibres increased from 2.2 nm to 2.9 nm with the molar ratio of acrylamide and GK₁₁ increasing

from 16.3 to 32.6. This indicates that the polyacrylamide hidden length was indeed coated on the peptide fibres and longer hidden length led to thicker polyacrylamide layer on the peptide fibres.

Then the SMFS of metal ion-clad picot fibres was studied. As show in Supplementary Fig. 6a, the folded polymers were picked up from random positions along their contour by the cantilever and stretched to mechanically unfold the peptide structures and release the polyacrylamide hidden length. This gives rise to sawtooth-like traces (colored in blue) with each peak corresponding to the rupture of the interactions between a pair of peptide strands (Supplementary Fig. 6b and c). These peaks can be adequately fitted using worm-like chain (WLC) models (red lines). The contour length increment of each peak (ΔL) was considered as the hidden length between peptides. With the molar ratio of acrylamide and GK₁₁ increasing from 16.3 to 32.6, the average hidden length increased from 5.4 to 9.5 nm (Supplementary Fig. 6d and e). The measured values of the hidden length were close to the theoretical values of 4.9 nm and 9.8 nm, further revealing that most of the acrylamide were integrated to the hidden length of picot fibres. The new results and discussion were included in the revised manuscript and Supplementary Information. (See the second paragraph on Page 10 in the revised manuscript; See the section of “AFM based single molecule force spectroscopy (SMFS) measurements” on Page 4 and Supplementary Fig. 5-6 on Page 12-13 in the revised Supplementary Information)

Revisions:

...Driven by the self-assembly of the peptides, the links between peptides are zipped as picots to contribute to the hidden length (Fig. 1b). The picots are at different lengths and the peptides connected with each other randomly...

...Here we present a type of hydrogels comprising hierarchical structures of picot fibres made of copper-bound self-assembling peptide strands with zipped flexible hidden length. The redundant hidden lengths allow the fibres to be extended to dissipate

mechanical load without reducing network connectivity, making the hydrogels robust against damage...

...Inspired by the structures of biological networks, we propose a new type of hierarchical structure made of picot fibres (p-fibres) consisting of self-assembled metal ion-clad peptide β -strands interconnected by flexible hidden lengths...

Furthermore, the hidden length in picot fibres can be tuned by adjusting the acrylamide concentration. As shown in Supplementary Fig. 5, the metal ion-clad picot fibres became thicker at the higher ratio of acrylamide and peptide, indicating that increasing ratio of polyacrylamide led to the longer hidden length. This was also confirmed by the AFM based single molecule force spectroscopy (SMFS) of metal ion-clad picot fibres (Supplementary Fig. 6). The hidden length increased from ~ 5.4 nm to ~ 9.5 nm when the molar ratio of acrylamide and ACLT-GK₁₁ increased from 16.3 to 32.6. The measured values of the hidden length were close to theoretical values of 4.9 nm and 9.8 nm, further revealing that most of the acrylamide was integrated to the hidden length of picot fibres.

AFM based single molecule force spectroscopy (SMFS) measurements

For the preparation of cantilevers used in SMFS, silicon nitride (Si₃N₄) cantilevers (MLCT, Bruker, USA) were first cleaned with Milli-Q water, and then placed in a chromic mixture (chromic acid) at 80 °C for 30 min. After that, the cantilevers were washed with deionized water, then ethanol, and dried under a steam of nitrogen. For the preparation of metal ion-clad picot fibres absorbed on glass substrates used in SMFS, ACLT-GK₁₁ (43.2 mM) and acrylamide (703 mM or 1406 mM) were copolymerized in deionized water and diluted for 3 times using 1 M Tris buffer (pH=7.60, containing 300

mM KCl and 150 mM CuCl₂) to induce self-assembly and ion binding. For the preparation of substrates, the glass substrates were cut into 1 × 1 cm² slides and soaked in a freshly prepared chromic mixture overnight. Then the substrates were washed with deionized water and ethanol, and then dried under a stream of nitrogen. The as prepared ion-clad picot fibre solution (50 μL) was dropped on the glass substrate and allowed to absorb for 30 min. The solution was removed, and the substrate was washed using Tris buffer for 3 times to remove the unabsorbed fibres.

AFM force spectroscopy experiments were carried out on a commercial AFM (JPK Nanowizard II). The force–distance curves were recorded by commercial software from JPK and analyzed by custom-written procedures in Igor pro 6.37 (Wavemetrics, Inc.). All the experiments were conducted at the room temperature (22 °C) and performed in 1 M Tris buffer (pH=7.60, containing 300 mM KCl). Soft silicon nitride MLCT-D cantilevers with typical spring constants of 30–45 pN nm⁻¹ were used for all experiments and calibrated using the thermal tune method. Typically, the cantilever was brought in contact with the substrate and held at the surface for 3 s, then retracted at a constant velocity of 2.0 μm s⁻¹. The sampling rate was 8 kHz. The contour length increment was determined by fitting the force peaks using the worm-like chain model with persistence lengths in a range of ~0.2–0.4 nm. The contour length increment of each peak (ΔL) was considered as the hidden length between peptides.

Fig. 1. Hydrogel design based on metal ion-clad picot fibres. **a**, Schematic of a conventional double-network hydrogel. The physically crosslinked network is designed to dissipate energy, and the covalently crosslinked network is used to provide hidden length. **b**, Schematic of the hydrogel constructed by picot fibres made of self-assembling peptide strands with zipped flexible **hidden lengths**. Upon deformation of the hydrogel, the picot fibres are extended to dissipate energy efficiently and release the polyacrylamide hidden length without reducing the network connectivity of the hydrogels. **c**, The peptide sequence and synthetic scheme for the metal ion-clad picot fibres. **d**, The self-assembled structure of GK₁₁ (left) and a magnified region of this diagram showing the binding position of Cu²⁺ (right). The hydrogen bonds are shown as grey dashed lines. **Histidine is coloured in green in the ribbon representation in the**

left image and represented by sticks to illustrate the coordination bonds in the right image.

Supplementary Figure 5. AFM imaging of metal ion-clad picot fibres ($\text{pGK}_{11}/\text{Cu}^{2+}$) at different acrylamide:ACLT-GK₁₁ ratios. a, AFM image (left) and height profile (right) of metal ion-clad picot fibres ($\text{pGK}_{11}/\text{Cu}^{2+}$) at the acrylamide:ACLT-GK₁₁ ratio of 16.3. The height profile at right corresponds to the cross line (green) in the left image. Scale bar = 1 μm. b, Diameter distribution of metal ion-clad picot fibres ($\text{pGK}_{11}/\text{Cu}^{2+}$) at the acrylamide:ACLT-GK₁₁ ratio of 16.3. c, AFM image (left) and height profile (right) of metal ion-clad picot fibres ($\text{pGK}_{11}/\text{Cu}^{2+}$) at the acrylamide:ACLT-GK₁₁ ratio of 32.6. The height profile at right corresponds to the cross line (green) in the left image. Scale bar = 1 μm. d, Diameter distribution of metal ion-clad picot fibres ($\text{pGK}_{11}/\text{Cu}^{2+}$) at the acrylamide:ACLT-GK₁₁ ratio of 32.6.

Supplementary Figure 6. SMFS of metal ion-clad picot fibres at different ratios of acrylamide and ACLT-GK₁₁. a, Schematic of the AFM-based SMFS for metal ion-clad picot fibres. b, c, Representative force–distance curves for the rupture of the metal ion-clad picot fibre at acrylamide:ACLT-GK₁₁ ratios of 16.3 (b) and 32.6 (c). The pulling speed was $2 \mu\text{m s}^{-1}$ and each peak corresponds to an individual rupture event between a pair of peptide strands. Red lines correspond to worm-like chain (WLC) fitting to the rupture events using the persistence length of 0.2-0.4 nm. ΔL indicates the hidden length between fractured peptides. d, e, Histograms of hidden lengths in metal ion-clad picot fibres at the acrylamide:ACLT-GK₁₁ ratio of 16.3 (d) and 32.6 (e). The measured hidden lengths (ΔL) were 5.4 nm (N=230) and 9.5 nm (N=316), respectively. Red lines correspond to a Gaussian fit.

Supplementary Figure 11. Schematic of the step-by-step release of the hidden length upon rupture of neighbouring β strands in the picot fibres.

2. According to the proposed structure, mechanical properties of the p-Pep/Cu²⁺ hydrogel should be dominated by enthalpic contribution of inter-peptide hydrogen bonding, not by entropic contribution. Thus, analysis of the elasticity based on the classical rubber elasticity is not suitable.

Response: We thank the reviewer for the comments. The reviewer is correct that “the mechanical properties of the p-Pep/Cu²⁺ hydrogel should be dominated by enthalpic contribution of inter-peptide hydrogen bonding, not by entropic contribution”. At low strains, the peptide fibres remain integrated without unfolding. Therefore, it is possible to estimate the elasticity of the hydrogels at this region based on rubber elasticity and considering the peptide fibres as unextendible rods. Indeed, the calculations based on rubber elasticity matched with our experimentally obtained values. At high strains, the elasticity of the hydrogels can be affected by the rupture of the peptide fibres as well as the alignment of the fibres to the force direction, which is quite complicated. Note that, the unfolding of the peptide fibres contributes largely to the fracture strain and toughness of the hydrogels. We have clarified this in the revised manuscript. (See the middle of Page 16 and the first paragraph on Page 17 in the revised manuscript)

Revisions:

...At low strains, the peptide fibres remain integrated without unfolding. Thus, the polyacrymide remains as the hidden length and does not contribute to the elasticity of the p-Pep/Cu²⁺ hydrogels. The hydrogels are formed by 4-armed PEG-ACLT with the picot fibres as unextendible crosslinkers. The elasticity of the hydrogels can be predicted using the classical rubber elasticity theory...

...It is worth noting that at high strains, the elasticity of the hydrogels can be affected by the rupture of the peptide fibres as well as the alignment of the fibres to the force direction, which is quite complicated. Moreover, the unfolding of the peptide fibres contributes largely to the fracture strain and toughness of the hydrogels.

3. As mentioned before, basically the p-Pep/Cu²⁺ hydrogel is a kind of dual cross-linking gels with physical and chemical bonds. In comparison with other gels having physical bonding, what is the key physical factor to obtain the good mechanical properties including fast recovery? The reviewer requests the authors to add more comments about this.

Response: We thank the reviewer for the comments. For hydrogels with normal network structures, stiffness is linearly proportional to the crosslinking density, and toughness is inversely proportional to the square root of the crosslinking density of hydrogels. Therefore, both chemical and physical crosslinks stiffen hydrogels with the trade-off of reducing their toughness. In contrast, in the p-Pep/Cu²⁺ hydrogels, the physical bonds were mainly used to build the metal ion-clad picot fibres. Each fibre can be considered as an extensible crosslinker. Increasing physical interactions by raising the peptide or metal ion concentrations only increases the total available sacrificial bonds in the fibres without affecting the crosslinking density of the hydrogels, thus resolving the stiffness-toughness trade-off common in polymer networks.

Besides, the unique hydrogel network structure formed by crosslinked picot fibres (Fig. 1b) also contributed to the fast mechanical recovery of the p-Pep/Cu²⁺ hydrogels. Rupture and reformation of the picot fibres mainly involved the recovery of

intramolecular interactions which was a localized process independent of the recovery for the overall hydrogel network. In contrast, in typical hydrogels with dual physical and covalent crosslinks, the randomly distributed physical crosslinks are prone to rebinding at incorrect positions. Fully recovering the mechanical properties of the hydrogels requires the break and reform of the misregistered physical crosslinks, thus slowing the recovery of hydrogels. Moreover, most Cu^{2+} remained attached to the peptides even the fibres were unfolded upon stretching the hydrogels. These Cu^{2+} ions may serve as the nuclei for the folding of picot fibres, as the fibres without Cu^{2+} showed reduced recovery rates. See the Response to the Q7 of Reviewer #1 for details. The new data and discussion have been included in the revised manuscript and Supplementary Information. (See the end of Page 14, the first paragraph on Page 15 and the end of Page 19 in the revised manuscript; See Supplementary Fig. 22 on Page 24 in the revised Supplementary Information)

Revisions:

...For hydrogels with normal network structures, stiffness is linearly proportional to the crosslinking density⁵¹, and toughness is inversely proportional to the square root of the crosslinking density.^{52, 53} Therefore, both chemical and physical crosslinks stiffen hydrogels with the trade-off of reducing their toughness. In contrast, in the p-Pep/ Cu^{2+} hydrogels, the physical bonds were mainly used to build the metal ion-clad picot fibres. Each fibre can be considered as an extensible crosslinker. Increasing physical interactions by raising the peptide or metal ion concentrations only increases the total available sacrificial bonds in the fibres without affecting the crosslinking density of the hydrogels, thus resolving the stiffness-toughness trade-off common in polymer networks...

...Moreover, the p-Pep/ Cu^{2+} hydrogels recovered even faster than the p-Pep hydrogels. It is probably because Cu^{2+} remained attached to the unfolded picot fibres and thus facilitated reformation of the picot fibres in the unloaded hydrogels, as indicated by the almost unchanged mechanical strength and negligible Cu^{2+} release of p-Pep/ Cu^{2+}

hydrogels under stretching-relaxation cycles in Tris buffer (Supplementary Fig. 22). These Cu^{2+} ions may serve as the nuclei for the folding of picot fibres, as the fibres without Cu^{2+} showed reduced recovery rates.

Supplementary Figure 22. Mechanical property and Cu^{2+} release of p-Pep/ Cu^{2+} hydrogels under stretching-relaxation cycles in Tris buffer (pH=7.60, containing 300 mM KCl). a, Typical stress–strain curves for p-Pep and p-Pep/ Cu^{2+} hydrogels under stretching-relaxation cycles. The cycle numbers are 1, 10, 20, 40, 60, 80 and 100. b, Summary of the maximum stress and energy dissipation for p-Pep/ Cu^{2+} hydrogels under stretching-relaxation cycles. c, UV-vis spectra for mixture of sodium diethyldithiocarbamate (DDTC-Na) and Cu^{2+} at different concentrations of Cu^{2+} (1-50 μM). The concentration of DDTC-Na was 250 μM . d, Calibration curve of Cu^{2+} concentrations and $\text{OD}_{452\text{nm}}$ (UV-vis absorbance at 452 nm). e, UV-vis spectra of the hydrogel leachates after different cycles in the presence of DDTC-Na. f, Accumulated

percentage of released Cu^{2+} from the p-Pep/ Cu^{2+} hydrogels after different cycles. For b, d and f, values represent the mean and standard deviation (n=3 independent experiments).

4. For fig. S16, it is not clear how the authors estimate the theoretical value of 2.0. Please show the estimation process. Is it calculated based on the swelling ratio difference?

Response: We thank the reviewer for the comments. The calculation of the theoretical values of the average length between two neighbouring cross-linking points after and before the release of the hidden length was shown in Supplementary Fig. 16 (now has been changed into Supplementary Fig. 18). Considering the network of the p-Pep/ Cu^{2+} hydrogel, the average length between two neighbouring cross-linking points before the release of the hidden length can be determined as Eq. (2).

$$L_1 = L_{\text{acrylamide+pep}} + 2R_{PEG} \quad (2)$$

where $L_{\text{acrylamide+pep}}$ is the length of the picot fibre between two neighbouring cross-linking points and R_{PEG} is the mean square end-to-end distance of a linear PEG (5 kDa). $L_{\text{acrylamide+pep}}$ can be determined as $\frac{C_{\text{pep}} \times (W_{\text{pep}} + L_{\text{gap}})}{4C_{PEG}}$, in which C_{pep} is the molar concentration of ACLT-GK₁₁ peptide, W_{pep} is the average width of an ACLT-GK₁₁ peptide (width of a β sheet), L_{gap} is the average length of the gap between adjacent peptide in the assembled GK₁₁, and C_{PEG} is the molar concentration of 4-armed PEG-ACLT. Following the worm-like chain (WLC) model, R_{PEG} is determined as $R_{PEG} = \sqrt{2\zeta_{PEG}L_{PEG}}$, in which ζ_{PEG} and L_{PEG} are the persistence length and the contour length of PEG, respectively. We used the ζ_{PEG} of 0.38 nm and L_{PEG} of 40 nm according to the previous literatures. The average width of a peptide (width of the β sheet) was 0.35 nm and the gap between adjacent peptide was 0.30 nm according to the simulation.

The average length between two neighbouring cross-linking points after the release of the hidden length can be determined as Eq. (3).

$$L_2 = R_{\text{acrylamide+pep}} + 2R_{\text{PEG}} \quad (3)$$

in which $R_{\text{acrylamide+pep}}$ is the mean square end-to-end distance of copolymerized acrylamide and peptide, R_{PEG} is the mean square end-to-end distance of PEG. The mean square end-to-end distance of copolymerized acrylamide and peptide can be determined following the WLC model.

$$R_{\text{acrylamide+pep}} = \sqrt{2 \zeta_{\text{acrylamide+pep}} L_{\text{acrylamide+pep}}} \quad (4)$$

where $\zeta_{\text{acrylamide+pep}}$ and $L_{\text{acrylamide+pep}}$ are the persistence length and contour length of copolymerized acrylamide and peptide, respectively. The persistence length of the copolymer of acrylamide and peptide was taken as 0.4 nm. The contour length of copolymerized acrylamide and peptide can be determined as Eq. (5).

$$L_{\text{acrylamide+pep}} = \frac{C_{\text{acrylamide}} + C_{\text{pep}}}{4C_{\text{PEG}}} \times 2L_{\text{C-C}} \quad (5)$$

where $L_{\text{C-C}}$ is the length of the C-C bond, $C_{\text{acrylamide}}$, C_{pep} and C_{PEG} are the molar concentrations of acrylamide, ACLT-GK₁₁ and PEG. The length of C-C bond was considered as 0.15 nm. Finally, the theoretical ratio of the average length between two neighbouring cross-linking points after and before the release of the hidden length was calculated as $\frac{L_2}{L_1} = 2.0$. The details about the estimation of the theoretical value of 2.0 is included in the section of ‘‘Calculation of the hidden length in the hydrogel network’’ in the Supplementary Information. (See the section of ‘‘Calculation of the hidden length in the hydrogel network’’ on Page 7-9 in the revised Supplementary Information)

Revisions:

Calculation of the hidden length in the hydrogel network

Considering the network of the p-Pep/Cu²⁺ hydrogel, the average length between two neighbouring cross-linking points before the release of the hidden length (top of Supplementary Fig. 18) can be determined as Eq. (2).

$$L_1 = L_{\text{acrylamide+pep}} + 2R_{\text{PEG}} \quad (2)$$

where $L_{\text{acrylamide+pep}}$ is the length of the picot fibre between two neighbouring cross-

linking points and R_{PEG} is the mean square end-to-end distance of a linear PEG (5 kDa). $L_{acrylamide+pep}$ can be determined as $\frac{C_{pep} \times (W_{pep} + L_{gap})}{4C_{PEG}}$, in which C_{pep} is the molar concentration of ACLT-GK₁₁ peptide, W_{pep} is the average width of an ACLT-GK₁₁ peptide (width of a β sheet), L_{gap} is the average length of the gap between adjacent peptide in the assembled GK₁₁, and C_{PEG} is the molar concentration of 4-armed PEG-ACLT. Following the worm-like chain (WLC) model, R_{PEG} is determined as $R_{PEG} = \sqrt{2\zeta_{PEG}L_{PEG}}$, in which ζ_{PEG} and L_{PEG} are the persistence length and the contour length of PEG, respectively. We used the ζ_{PEG} of 0.38 nm and L_{PEG} of 40 nm according to the previous literatures.¹⁵⁻¹⁷ The average width of a peptide (width of the β sheet) was set as 0.35 nm and the gap between adjacent peptide was 0.30 nm according to the simulation.

The average length between two neighbouring cross-linking points after the release of the hidden length (bottom of Supplementary Fig. 18) can be determined as Eq. (3).

$$L_2 = R_{acrylamide+pep} + 2R_{PEG} \quad (3)$$

in which $R_{acrylamide+pep}$ is the mean square end-to-end distance of copolymerized acrylamide and peptide, R_{PEG} is the mean square end-to-end distance of PEG. The mean square end-to-end distance of copolymerized acrylamide and peptide can be determined following the WLC model.

$$R_{acrylamide+pep} = \sqrt{2\zeta_{acrylamide+pep}L_{acrylamide+pep}} \quad (4)$$

where $\zeta_{acrylamide+pep}$ and $L_{acrylamide+pep}$ are the persistence length and contour length of copolymerized acrylamide and peptide, respectively. 0.4 nm was used as the persistence length of copolymerized acrylamide and peptide.¹⁸ The contour length of copolymerized acrylamide and peptide can be determined as Eq. (5).

$$L_{acrylamide+pep} = \frac{C_{acrylamide} + C_{pep}}{4C_{PEG}} \times 2L_{C-C} \quad (5)$$

where L_{C-C} is the length of the C-C bond, $C_{acrylamide}$, C_{pep} and C_{PEG} are the molar concentrations of acrylamide, ACLT-GK₁₁ and PEG. The length of C-C bond was considered as 0.15 nm. Finally, the theoretical ratio of the average length between two

neighbouring cross-linking points after and before the release of the hidden length was calculated as $\frac{L_2}{L_1} = 2.0$.

Based on the classical rubber elasticity theory and Gaussian chain statistics¹⁹⁻²², the stress of the hydrogel can be described as Eq. (6).

$$\sigma = G \left(\lambda - \frac{1}{\lambda^2} \right) = \frac{\rho RT}{M_c} \left(\lambda - \frac{1}{\lambda^2} \right) \quad (6)$$

in which σ is the stress, $G = \frac{\rho RT}{M_c}$ is the shear modulus, ρ is the density of the polymer in the network, M_c is the molecular weight between two neighbouring cross-linking points, R is the gas constant, and λ is the extension ratio. Young's modulus E is defined as $E = \frac{\partial \sigma}{\partial \lambda}$ and $E = 3G$ when $\lambda = 1$. The ratio of Young's modulus for the same hydrogel before and after the release of the hidden length is $\varepsilon = \frac{\rho_1 M_{c2}}{\rho_2 M_{c1}}$. According to the Young's modulus and swelling ratio of the hydrogel before and after the treatment of GuHCl in experiments, the ratio of M_c after and before the release of the hidden length is $\sim 1.8 \pm 0.3$. We assume that the length of the polymer chain in the hydrogel network is directly proportional to its molecular weight. As a result, the ratio of the average length between two neighbouring cross-linking points after and before the release of the hidden length in experiments was calculated as $\sim 1.8 \pm 0.3$.

5. "As these sacrificial bonds cannot reform efficiently, the hydrogels lack a mechanism to prevent crack propagation and are susceptible to fatigue under cyclic loading"; the reviewer is unsure of the relationship between them. Is the slow reformation of sacrificial bonds really the reason of the poor fatigue resistance?

Response: We thank the reviewer for the comments. Studies by Lake and coworkers on rubbers (*Rubber Chem. Technol.* 1995, 68, 435–460; *Proc. R. Soc. Lond. A Math. Phys. Sci.* 1967, 300, 108–119) and Suo and coworkers on hydrogels (*Eur. J. Mech. A, Solids* 2019, 74, 337–370, *ACS Macro Lett.* 2018, 7, 312–317, *ACS Macro Lett.* 2019, 8, 17–23) suggested fatigue threshold is determined by the covalent network, whereas the noncovalent/reversible interactions make negligible contributions. However, this role cannot quantitatively match the fatigue threshold of soft materials containing

dynamic bonds. Recently, Gong et. al. studied the role of dynamic bonds on fatigue threshold of tough hydrogels (*Proc. Natl. Acad. Sci. USA* 2022, 119 (20), e2200678119.). They raised the point that in the viscoelastic regime, the dynamic bonds carry load, protecting the permanent bonds from breaking, and thereby contribute to the fatigue threshold. The fatigue resistance of soft materials can be improved by designing dynamic bonds with a relatively long relaxation time so that the observation (working) time window falls in the viscoelastic regime. As a result, the slow reformation of sacrificial bonds after fracture cannot provide long-term protections on the permanent bonds in cyclic loading, thus leading to the poor fatigue resistance. Moreover, the short relaxation time of sacrificial bonds with slow reformation rates prevent the observation (working) time window falling in the viscoelastic regime, thus decreasing the fatigue resistance. Now we have included the new comments and cited the new reference in the revised manuscript. (See the bottom of Page 3, the end of Page 20, the first paragraph on Page 21 and reference No. 30 in the revised manuscript)

Revisions:

...As these sacrificial bonds cannot reform efficiently, the hydrogels lack a mechanism to prevent crack propagation and are susceptible to fatigue under cyclic loading³⁰...

...A recent study showed that dynamic interactions can only carry load and contribute to the fatigue resistance when their lifetimes are longer than the timescale of the periodic mechanical load.³⁰ Note that both the lifetimes and recovery rates of typical physical interactions are considerably reduced by applied force.^{64, 65} Therefore, they cannot carry load under cyclic stretch to prevent crack propagation. We attribute the outstanding fatigue resistance of the p-Pep/Cu²⁺ hydrogel to the high mechanical stability and fast reassembly of the metal ion-clad picot fibres. In addition, the release of hidden length can also prevent crack propagation and increase the fatigue threshold according to the Lake-Thomas model^{52, 66}...

6. "it remains challenging to achieve high strength, high toughness, robust fatigue

resistance, and rapid recovery in the same synthetic hydrogel”; please check the two recent papers published in Science, Liu et al., Science 2021, 372, 1078 / Kim et al., Science 2021, 374, 212. These works have realized high toughness of elastic hydrogels (with rapid recovery) with different strategies.

Response: We thank the reviewer for the comments. We agree with the reviewer that Liu et al. and Kim et al. presented different strategies to realize high toughness of elastic hydrogels (with rapid recovery). Liu et al. used the damageless reinforcement strategy of strain-induced crystallization to build high toughness and rapid recovery. Kim et al. used dense entanglements which greatly outnumber cross-links in hydrogels to realize high toughness, strength, and fatigue resistance. However, they did not combine all the high strength, high toughness, robust fatigue resistance, and rapid recovery in the same synthetic hydrogel, which can be addressed to a certain extent using the design of picot fibres in our work. We have cited these works and added comments in the revised manuscript. (See the end of Page 3, the first paragraph on Page 4 and reference No. 31-32 in the revised manuscript)

Revisions:

...Recently, some efforts have partially overcome the trade-offs among these mechanical properties using strain-induced self-reinforcement³¹, chain entanglement³² or mechanochemically activable hidden length³³. For example, Liu et al. used the damageless reinforcement strategy of strain-induced crystallization to build high toughness and rapid recovery.³¹ Kim et al. used dense entanglements which greatly outnumber cross-links in hydrogels to realize high toughness, strength, and fatigue resistance.³² Moreover, the hidden length in fibril structures or folded proteins can provide extra extensibility after being released, thus enhancing the toughness of hydrogels.³³⁻³⁶...

7. The p-Pep/Cu²⁺ hydrogel is more deformable than the PAM gel. While the authors assume that this is due to formation of the peptide fibers, it is also possible that lower swelling ratio of the p-Pep/Cu²⁺ gel ensures its large deformability because swelling of

a gel generally decreases its deformability.

Response: We thank the reviewer for the comments. We agree with the reviewer that the lower swelling ratio of the p-Pep/Cu²⁺ hydrogel would lead to the large deformability. Actually, we think that the large deformability due to lower swelling ratio is consistent with the formation of the peptide fibres. Since GK₁₁ peptide can form fibres in hydrogels, the acrylamide was zipped as the hidden length of the peptide fibres. Hence, acrylamide would not contribute to the swelling of hydrogels before the release of hidden length, leading to the low swelling ratio of p-Pep/Cu²⁺ hydrogels. During the deformation, fracturing the peptide fibres can release the hidden length without reducing network connectivity of the hydrogels, leading to the significantly enhanced deformability. This is further confirmed by the increased fracture strain of hydrogels at higher acrylamide concentrations (Supplementary Fig. 15c-d). Now, we have included the discussion in the revised manuscript. (See the second paragraph on Page 12 in the revised manuscript and Supplementary Fig. 15 on Page 17 in the revised Supplementary Information)

Revisions:

...The deformability of p-Pep/Cu²⁺ hydrogels can be mainly attributed to the zipped acrylamide in hidden length of the peptide fibres, which also contributed to the low swelling ratio. The released hidden length of peptide fibres during deformations led to the significantly enhanced deformability...

Supplementary Figure 15. Mechanical properties of the p-Pep/Cu²⁺ hydrogels with various 4-armed PEG-ACLT or acrylamide concentrations. a, Uniaxial stretching stress–strain curves of p-Pep/Cu²⁺ hydrogels with different 4-armed PEG-ACLT concentrations (0%, 1%, 3%, 5% and 7% w/v) in the precursors during preparation. b, Summaries of Young’s modulus and **work of rupture** of p-Pep/Cu²⁺ hydrogels at different 4-armed PEG-ACLT concentrations (0%, 1%, 3%, 5% and 7% w/v) in the precursors during preparation. c, Uniaxial stretching stress–strain curves of p-Pep/Cu²⁺ hydrogels at different acrylamide concentrations in the precursors during preparation (225, 325 and 450 mg mL⁻¹). d, Summaries of Young’s modulus and **work of rupture** of p-Pep/Cu²⁺ hydrogels at different acrylamide concentrations in the precursors during preparation (225, 325 and 450 mg mL⁻¹). For b and d, values represent the mean and standard deviation (n = 3 independent samples).

8. About fig. S8, use of SEM to estimate mesh size of a wet gel is generally not suitable. Such a “mesh” in a dried gel is probably an artifact and formed during a drying process.
Response: We thank the reviewer for the comments. We totally agree with the reviewer that SEM cannot represent the precise mesh size of a wet hydrogel quantitatively. SEM images can only be used to distinguish the mesh size of different hydrogels qualitatively, because the original mesh structures cannot be fully maintained in the absence of the solvent after lyophilization. To avoid confusion, now we have removed the SEM images

and the statement of mesh size in the revised manuscript. (See the end of Page 11 in the revised manuscript)

Revisions:

~~... Scanning electron microscopy (SEM) confirmed that the picot fibre hydrogels had smaller mesh sizes than the PAM hydrogels and that the mesh size further decreased with ion coordination (Supplementary Fig. 8). All these results indicate the hydrogel network structures are affected by the formation of picot fibres and metal ion binding.~~

9. Please define the term “toughness” carefully. The authors used the term “toughness” for two different physical quantities. One is the area below the tension stress-strain curve until fracture with the unit of J m^{-3} . The other is the fracture energy with the unit of J m^{-2} , which is called “toughness” in the introduction and conclusion. While the latter should be physically called as toughness, in this field both values are often called as toughness. The reviewer suggests using a different name for the former instead of “toughness” to avoid confusion.

Regarding to this, the authors wrote “toughness is inversely proportional to the square root of the crosslinking density”. This “toughness” is the latter one, not the former one.

Response: We thank the reviewer for the insightful comments. We agree with the reviewer that the term “toughness” should be defined clearly in the manuscript. Following the reviewer’s suggestion, the area below the tension stress-strain curve until fracture with the unit of J m^{-3} is defined as the work of rupture (W_r) (*Liu et al., Science 2021, 372, 1078*). The fracture energy with the unit of J m^{-2} is defined as toughness. The terms have been revised throughout the manuscript and Supplementary Information. (See the end of Page 13, the second paragraph on Page 14, the second paragraph on Page 15, the second paragraph on Page 25 and Fig. 4 on Page 17 in the revised manuscript; See Supplementary Fig. 15-16 on Page 17-18 and Supplementary Table 1-2 on Page 25-26 in the revised Supplementary Information)

Revisions:

...The fracture stress, Young's modulus and the work of rupture (W_r) for the p-Pep/Cu²⁺ hydrogel were significantly higher than those of the PAM and p-Pep hydrogels (Fig. 4a and Supplementary Table 1)...

...The work of rupture of PAM, p-Pep and p-Pep/Cu²⁺ hydrogels were ~0.4, 7.3 and 38.1 MJ m⁻³, respectively...

...The fracture stress and work of rupture also increased at increasing GK₁₁ concentrations at low peptide concentrations up to 6% and dropped when further increasing the peptide, presumably due to the aggregation of the peptide fibres (Supplementary Fig. 12)...

...The fracture strain and work of rupture of the p-Pep/Cu²⁺ hydrogels reached more than 2500% and 50 MJ m⁻³ by simply increasing the concentration of acrylamide (Supplementary Fig. 15c-d). Work of rupture and strength of hydrogels were also enhanced by the picot fibres in the compressive mechanical tests (Supplementary Fig. 16)...

...The work of rupture (W_r) was calculated from the area below the tension stress-strain curve until fracture...

Fig. 4. Mechanical properties of hydrogels made of metal ion-clad picot fibres. **a**, Typical stress–strain curves under tension for PAM, p-Pep and p-Pep/Cu²⁺ hydrogels. **b**, Typical strain–stress curves of p-Pep/Cu²⁺ hydrogels under tension at various peptide concentrations in the precursor of hydrogels (0%, 2%, 4%, 6%, and 9% w/v). The inset corresponds to the summarized Young’s modulus and **work of rupture (W_r)** of the p-Pep/Cu²⁺ hydrogels. Values represent the mean and standard deviation ($n=3$ independent samples). **c**, Typical stretching–relaxation cycles of PAM, p-Pep and p-Pep/Cu²⁺ hydrogels at a strain of 4 mm mm⁻¹. **d**, Stretching–relaxation cycles for p-Pep/Cu²⁺ hydrogels subjected to various strains (4, 8, 12, 16, and 20 mm mm⁻¹). The curves are offset for clarity, and the overlapping curves are shown in the inset image. **e**, Stress–strain curves for 100 consecutive stretching–relaxation cycles without any stops for p-Pep and p-Pep/Cu²⁺ hydrogels. The cycle numbers are 1, 10, 20, 30, 40, 50, 60, 70, 80, 90 and 100. **f**, Extension of cracks per cycle as a function of the energy release rate for the PAM, p-Pep and p-Pep/Cu²⁺ hydrogels. **The R^2 values of the linear fitting to estimate fatigue thresholds for PAM, p-Pep and p-Pep/Cu²⁺ hydrogels were 0.99, 0.99 and 0.95, respectively. Values represent the mean and standard deviation ($n=3$ independent experiments).** For **a**, **c**, **d**, **e**, and **f**, the peptide concentration in the hydrogel precursors was 6% w/v.

Supplementary Figure 15. Mechanical properties of the p-Pep/Cu²⁺ hydrogels with various 4-armed PEG-ACLT or acrylamide concentrations. a, Uniaxial stretching stress–strain curves of p-Pep/Cu²⁺ hydrogels with different 4-armed PEG-ACLT concentrations (0%, 1%, 3%, 5% and 7% w/v) in the precursors during preparation. b, Summaries of Young's modulus and **work of rupture** of p-Pep/Cu²⁺ hydrogels at different 4-armed PEG-ACLT concentrations (0%, 1%, 3%, 5% and 7% w/v) in the precursors during preparation. c, Uniaxial stretching stress–strain curves of p-Pep/Cu²⁺ hydrogels at different acrylamide concentrations in the precursors during preparation (225, 325 and 450 mg mL⁻¹). d, Summaries of Young's modulus and **work of rupture** of p-Pep/Cu²⁺ hydrogels at different acrylamide concentrations in the precursors during preparation (225, 325 and 450 mg mL⁻¹). For b and d, values represent the mean and standard deviation (n = 3 independent samples).

Supplementary Figure 16. Mechanical properties of p-Pep/Cu²⁺ hydrogels under compression. a, Typical stress–strain curves under compression for PAM, p-Pep and p-Pep/Cu²⁺ hydrogels. b, Summaries of fracture strain and stress of the PAM, p-Pep and p-Pep/Cu²⁺ hydrogels under compression. c, Summaries of Young’s modulus and **work of rupture** of the PAM, p-Pep and p-Pep/Cu²⁺ hydrogels under compression. d, Typical stress–strain curves of p-Pep/Cu²⁺ hydrogels at various peptide concentrations in the precursor of hydrogels (0%, 2%, 4%, and 6% w/v) under compression. e, Summaries of fracture strain and stress of p-Pep/Cu²⁺ hydrogels at various peptide concentrations in the precursor of hydrogels (0%, 2%, 4%, and 6% w/v) under compression. f, Summaries of Young’s modulus and **work of rupture** of p-Pep/Cu²⁺ hydrogels at various peptide concentrations in the precursor of hydrogels (0%, 2%, 4%, and 6% w/v) under compression. For b, c, e, and f, values represent the mean and standard deviation (n = 3 independent samples).

Supplementary Table 1. Tensile mechanical properties of PAM, p-Pep and p-Pep/Cu²⁺ hydrogels containing different concentrations of peptides. Values represent the mean and standard deviation.

Peptide concentration n (v/w)	Fracture strain (mm mm ⁻¹)	Fracture stress (MPa)	Young’s modulus (kPa)	Work of rupture (MJ m ⁻³)	Toughness (kJ m ⁻²)
PAM /	5.4 ± 0.8	0.18 ± 0.03	13.2 ± 3.2	0.4 ± 0.04	0.10 ± 0.01

p-Pep	6%	14.1 ± 0.9	1.35 ± 0.12	25.4 ± 5.9	7.3 ± 1.6	2.6 ± 0.6
	2%	7.6 ± 0.3	0.40 ± 0.03	20.9 ± 3.7	1.3 ± 0.4	--
p-Pep/Cu ²⁺	4%	16.8 ± 0.7	1.92 ± 0.17	29.5 ± 2.8	12.6 ± 1.6	--
	6%	22.5 ± 1.4	4.12 ± 0.37	71.5 ± 4.6	38.1 ± 3.6	25.3 ± 1.9
	9%	18.2 ± 1.2	3.66 ± 0.21	121.5 ± 14.0	29.1 ± 3.5	--

Supplementary Table 2. Mechanical properties of the p-Pep/Cu²⁺ hydrogel, and other tough and rapid recovery hydrogels from the literature.

Classification	Sample code	Water content (wt%)	Young's modulus (MPa)	Fracture strain (mm/mm)	Fracture stress (MPa)	Work of rupture (MJ m ⁻³)	Toughness (kJ m ⁻²)	Recovery time (min)	Recovery efficiency (%)	Ref. No.
Single-network hydrogel	p-Pep/Cu ²⁺	87	0.07	22.5	4.12	38.1	25.3	0	~100	This work
		82 (*)	0.1 (*)	25.0 (*)	5.47 (*)	55.7 (*)	--	--	--	
	Highly entangled hydrogel	~70	0.05	~4.5	0.4	--	2.2	--	--	23
	Polyampholytes gel	50-70	0.01-8	1.5-15	0.1-2	0.1-7.0	1.0-4.0	120	~100	24
	(FL) ₈ gel	70	0.016	4.5	0.035	--	--	20	~85	25
	DMAA-MAAc hydrogel	28	8	8	2	--	9.3	3 (37 °C)	~100	26
Dual-crosslinked hydrogel	P(urea-ILa-SPMA _s)-3d gel	~50	1.97	4.78	1.90	6.70	--	120	~85	27
	PIC gel	55	5.4	7.5	3.8	18.8	10.0	120	85	28
	CB[8] gel	90	0.0046	24	~130	--	0.75	3	~100	29
	HN-PH ₆	80.4	0.27	4.12	3.02	4.03	--	0	~85	30
	D-hydrogel-0.15	60-70	~1.75	7.48	5.9	27.2	--	240	87.6	31
	CCP-MCP1 gel	32	0.145	5.49	2.6	--	1.33	>15	--	32
Double-network hydrogel	Ca ²⁺ -alginate-PAAm	86	0.029	23	0.156	--	8.7	1440	74	14
	B-DN3 gel	44	22	5.7	10.5	--	2.85	5	~85	33
	Agar/PAMAAc-Fe ³⁺ DN	--	0.27	14	1.55	16.7	0.894	20	95	34
	DN-Sul gel	54.6	0.8	5.05	3.7	7.6	9.8	240	>90	35
	DN-Cit gel	56.9	1.3	5	5.6	12.1	14	240	96.6	35
	PAM-CS-S DN gel	~80	0.357	5.6	1.94	--	8.3	240	90	36
Nanocomposite hydrogel	L-NC gel	62	43.2	7.4	1.6	7.38	--	--	--	37
	SHARK hydrogel		0.03	77	1.02	32.6	19.75	--	--	38

(*) The concentration of acrylamide in the precursor of hydrogels was 450 mg mL⁻¹.

REVIEWERS' COMMENTS

Reviewer #1 (Remarks to the Author):

The authors addressed all my initial comments and it is acceptable for publication.

Reviewer #2 (Remarks to the Author):

The authors extensively modified the manuscript based on the reviewers' comments. Regarding the reviewer's concerns about gel structure, the authors additionally performed SMFS experiment and verified that single picot fibre with PAAm linker certainly exhibit hidden length effect. The reviewer now suggests publication of this manuscript.

Point-by-point response to the reviewers' comments

Reviewer #1 (Remarks to the Author):

Xue et al. have developed a novel hydrogel based of a peptide-polymer composite with coordinating metal ions. They created a hybrid bio-material with toughness and resistance to fracture. To increase the strength of the hydrogel, Cu^{2+} ions were added and coordinated between histidine residues between antiparallel peptide strands, resulting in a beta-sheet structure. The hydrogels were subjected to stress, fatigue, and hysteresis experiments.

Specific Comments:

1. Hidden length was not discussed in the introduction. A brief sentence or two explaining why hidden length is important for increasing toughness would be helpful.

Response: We thank the reviewer for the comments. Following the reviewer's suggestion, we have added a brief sentence explaining why hidden length is important for increasing toughness in the introduction. The hidden length in fibril structures or folded proteins can provide extra extensibility and increase the fracture strain after being released, thus greatly enhancing the toughness of hydrogels. The new comments have been included in the revised manuscript. (See the first paragraph on Page 4 in the revised manuscript)

Revisions:

...Moreover, the hidden length in fibril structures or folded proteins can provide extra extensibility after being released, thus enhancing the toughness of hydrogels.³³⁻³⁶...

2. In figure 1D, can you offer any rationale for why a second Cu^{2+} is not coordinated between 2 adjacent strands? Also, consider inverting the color scheme (i.e. green for histidines and white for other residues).

Response: We thank the reviewer for the comments. According to the structure

predicted by Alphafold2, the β -sheet of a single GK₁₁ peptide is slightly twisted ($\theta \sim 12^\circ$), making side chains of histidine (imidazole) distributing on both sides of the β -sheet in the structure of well-aligned anti-parallel β -sheets. Hence, histidine at the position of H₂, H₄ and H₆ were on the different side of the β -sheet with the histidine at the position of H₁₀ (Fig. 1d). Considering the tetrahedral coordination of Cu²⁺ and histidine, the second Cu²⁺ prefers to form coordination on the other side of the β -sheet with a third strand. Moreover, we also inverted the color scheme, and used green for histidine and white for other residues, following the reviewer's suggestion. (See the first paragraph and Fig. 1 on Page 7 in the revised manuscript)

Revisions:

...Noting that the β -sheet of a single GK₁₁ peptide is slightly twisted ($\theta \sim 12^\circ$), making side chains of histidine (imidazole) distributing on both sides of the β -sheet. Considering the tetrahedral coordination of Cu²⁺ and histidine, the second Cu²⁺ prefers to form coordination on the other side of the β -sheet with a third strand.

Fig. 1. Hydrogel design based on metal ion-clad picot fibres. **a**, Schematic of a conventional double-network hydrogel. The physically crosslinked network is designed to dissipate energy, and the covalently crosslinked network is used to provide hidden length. **b**, Schematic of the hydrogel constructed by picot fibres made of self-assembling peptide strands with zipped flexible **hidden lengths**. Upon deformation of the hydrogel, the picot fibres are extended to dissipate energy efficiently and release the polyacrylamide hidden length without reducing the network connectivity of the hydrogels. **c**, The peptide sequence and synthetic scheme for the metal ion-clad picot fibres. **d**, The self-assembled structure of GK₁₁ (left) and a magnified region of this diagram showing the binding position of Cu²⁺ (right). The hydrogen bonds are shown as grey dashed lines. **Histidine is coloured in green in the ribbon representation in the**

left image and represented by sticks to illustrate the coordination bonds in the right image.

3. At the end of section “Design of hydrogels based on metal ion-clad picot fibres”, the claim is made that the binding energy between the GK₁₁ peptide and Cu²⁺ was -7.2 kcal/mol and is five times the strength of a typical hydrogen bond in proteins. How is this measurement determined? Is it calculated computationally or is this experimentally determined. Moreover, is the ‘five times’ comparison being made a one-to-one comparison by experiment/simulation? The authors should add relevant citation(s) at the end of this claim.

Response: We thank the reviewer for the comments. The binding energy between the GK₁₁ peptide and Cu²⁺ was calculated based on the simulation. The GK₁₁-Cu²⁺ binding structure was first predicted using Alphafold2 neural network through ColabFold pipeline. Then the predicted GK₁₁-Cu²⁺ binding structure was used to calculate the binding free energy with the MM-PBSA method. For all trajectory with Amber15, the decomposition of binding free energy was calculated by considering the enthalpic part containing: Electrostatic, Van der Waals, Polar and Non-Polar Solvation.

$$\Delta G_{bind} = \Delta E_{ele} + \Delta G_{vdW} + \Delta G_{pol} + \Delta G_{nonpol} \quad (1)$$

We have indicated that the binding energy was determined by simulation in the revised manuscript and the simulation details are included in the section of Supplementary Methods in Supplementary Information. (See the first paragraph on Page 2 in the Supplementary Information)

The “five times” comparison is made based on the energy of hydrogen bond previously reported. According the studies of Sheh-Yi et al. and others (*P. Natl. Acad. Sci. USA* 2003, 100, 12683; *P. Natl. Acad. Sci. USA* 1993, 90, 1172), the bond energy of the hydrogen bond in water was ~1.5 kcal/mol, which is one fifth of the binding energy between the GK₁₁ peptide and Cu²⁺ in this work. We have added relevant citations at the end of this claim following the reviewer’s suggestion. (See the last paragraph on Page 6 and references No. 45-46 in the revised manuscript)

Revisions:

...The binding energy between the GK₁₁ peptide and Cu²⁺ was -7.2 kcal mol⁻¹ as determined by simulation, which is about five times the strength of a typical hydrogen bond in proteins^{45,46} ...

4. The limit of fracture stiffness is presumably due to aggregation of the peptide fibers. I think this is likely, but would a TEM show at low GK₁₁ and high GK₁₁ -concentrations confirm this?

Response: We thank the reviewer for the comments. In order to confirm the aggregation of the peptide fibers in hydrogels directly, we investigated the microstructures of p-Pep/Cu²⁺ hydrogels at different GK₁₁ concentrations (2% and 9% w/v) using field emission scanning electron microscope (FESEM) with a resolution of 0.8 nm. It has a similar resolution as TEM but can image hydrogel samples directly without the need of complicated sample slicing. As shown by Supplementary Fig. 12, the fibers aggregated into dense networks with increased entanglements at the high GK₁₁ concentration. The new results have now been included in the revised manuscript and Supplementary Information. (See the last paragraph on Page 14 in the revised manuscript; See the second paragraph on Page 5 and Supplementary Fig. 12 on Page 16 in the Supplementary Information)

Revisions:

...The fracture stress and work of rupture also increased at increasing GK₁₁ concentrations at low peptide concentrations up to 6% and dropped when further increasing the peptide, presumably due to the aggregation of the peptide fibres (Supplementary Fig. 12)...

Field emission scanning electron microscope (FESEM)

The FESEM images were obtained using a Quanta scanning electron microscope (GeminiSEM360, Zeiss, Germany) at 15 kV. The hydrogels were dialyzed in Milli-Q

water for 24 h to remove the unbound salts and lyophilized prior to the measurement.

Supplementary Figure 12. FESEM images of p-Pep/Cu²⁺ hydrogels at different GK₁₁ concentrations (2% and 9% w/v). a, FESEM images of p-Pep/Cu²⁺ hydrogels at the GK₁₁ concentration of 2% w/v. b, FESEM images of p-Pep/Cu²⁺ hydrogels at the GK₁₁ concentration of 9% w/v.

5. Please briefly describe how crack propagation was measured in addition to citing the work by Suo et al.

Response: We thank the reviewer for the comments. Generally, a hydrogel with a crack being created to the edge (20% of the width) was subjected to continuous stretching-relaxation cycles at the strain of λ . The hydrogel crack propagation was recorded using a camera from the undeformed hydrogels before and after the Nth cycle. Then the crack propagation was divided by the number of cycles to get the extension of crack per cycle, dc/dN , assuming that c linearly increases in N cycles. The details of the crack propagation determination were included in “Tensile and compressive test” of the Methods section in the manuscript. (See the last paragraph on Page 25 in the revised manuscript)

Revisions:

For the measurement of energy release rates, a hydrogel with a crack being created to

the edge was subjected to continuous stretching-relaxation cycles at the strain of λ . The crack propagation was recorded using a camera from the undeformed hydrogels before and after the Nth cycle. Then the crack propagation was divided by the number of cycles to get the extension of crack per cycle, dc/dN , assuming that c linearly increases in N cycles...

6. Can a standard deviation be applied to the “1.8 times the initial length” value given for the change in Young’s modulus before and after to make a more convincing closeness to the theoretical 2.0 value?

Response: We thank the reviewer for the comments. Considering the standard deviations of Young’s modulus and swelling ratios, the standard deviation of the ratio for the average length between cross-links after and before the release of the hidden length in experiments was 0.3. According to the calculation of the hidden length in the hydrogel network (See the section of “Calculation of the hidden length in the hydrogel network” on Page 7-9 in the Supplementary Information), the ratio of the average length between cross-links after and before the release of hidden length was calculated as $\varepsilon \frac{\rho_2}{\rho_1}$, in which ε is the ratio of Young’s modulus before and after the release of hidden length, ρ_2 and ρ_1 correspond to the densities of the polymer in the network before and after the release of hidden length. Considering the amount of polymer in the network remained consistent before and after the release of hidden length, the polymer density is inversely proportional to the swelling ratio of hydrogels before and after the addition of GuHCl. The standard deviation has now been applied to the “1.8 times the initial length” value in the revised manuscript. (See the end of Page 16 in the revised manuscript)

Revisions:

...The significant change in Young’s modulus before and after the addition of GuHCl indicated that the hidden length released between two neighbouring crosslinks in the hydrogels was 1.8 ± 0.3 times the initial length, which was close to the theoretical value

of 2.0 (Supplementary Fig. 18).

7. Is there any way to clarify if the faster recovery of Cu^{2+} bound p-Pep hydrogels is due to Cu^{2+} attached remnants?

Response: We thank the reviewer for the comments. Now we have performed experiments to show that most Cu^{2+} remained attached to the peptides during cyclic stretching. First, p-Pep/ Cu^{2+} hydrogels were subjected to stretching-relaxation cycles in 1 M Tris buffer (pH=7.60, containing 300 mM KCl) directly (Supplementary Fig. 22a-b). The maximum stress and energy dissipation of the hydrogel remained 93% and 88% in 100 cycles. The Cu^{2+} release after different cycles were detected using a Cu^{2+} sensor, sodium diethyldithiocarbamate (DDTC-Na) (Supplementary Fig. 22c-f). The total percentage of released Cu^{2+} was less than 0.5% in 100 stretching-relaxation cycles, indicating the negligible Cu^{2+} detachment (Supplementary Fig. 22f). Considering that the recovery of p-Pep/ Cu^{2+} hydrogels was much faster than that of the p-Pep hydrogels, these results suggested that the Cu^{2+} attached remnants contributed to the fast recovery of Cu^{2+} bound p-Pep hydrogels. (See the end of Page 19 in the revised manuscript; See the last paragraph on Page 6 and Supplementary Fig. 22 on Page 24 in the revised Supplementary Information)

Revisions:

Moreover, the p-Pep/ Cu^{2+} hydrogels recovered even faster than the p-Pep hydrogels. It is probably because Cu^{2+} remained attached to the unfolded picot fibres and thus facilitated reformation of the picot fibres in the unloaded hydrogels, as indicated by the almost unchanged mechanical strength and negligible Cu^{2+} release of p-Pep/ Cu^{2+} hydrogels under stretching-relaxation cycles in Tris buffer (Supplementary Fig. 22). These Cu^{2+} ions may serve as the nuclei for the folding of picot fibres, as the fibres without Cu^{2+} showed reduced recovery rates.

Cu^{2+} release from hydrogels

For the detecting of Cu^{2+} release from p-Pep/ Cu^{2+} hydrogels, a hydrogel was subjected

to stretching-relaxation cycles. After different cycles, the hydrogel was immersed into 1 M Tris buffer (pH=7.60, containing 300 mM KCl and 0.3 mM sodium diethyldithiocarbamate (DDTC-Na)) immediately for 30 min. The volume of Tris buffer was five times of the p-Pep hydrogel. Then the UV-vis spectra of the leachates were recorded using a V-550 (JASCO Inc., Japan) spectrophotometer. The absorbance at 452 nm was used to monitor the concentration of Cu^{2+} .

Supplementary Figure 22. Mechanical property and Cu^{2+} release of p-Pep/ Cu^{2+} hydrogels under stretching-relaxation cycles in Tris buffer (pH=7.60, containing 300 mM KCl). a, Typical stress-strain curves for p-Pep and p-Pep/ Cu^{2+} hydrogels under stretching-relaxation cycles. The cycle numbers are 1, 10, 20, 40, 60, 80 and 100. b, Summary of the maximum stress and energy dissipation for p-Pep/ Cu^{2+} hydrogels under stretching-relaxation cycles. c, UV-vis spectra for mixture of sodium diethyldithiocarbamate (DDTC-Na) and Cu^{2+} at different concentrations of Cu^{2+} (1-50

μM). The concentration of DDTC-Na was 250 μM . d, Calibration curve of Cu^{2+} concentrations and $\text{OD}_{452\text{nm}}$ (UV-vis absorbance at 452 nm). e, UV-vis spectra of the hydrogel leachates after different cycles in the presence of DDTC-Na. f, Accumulated percentage of released Cu^{2+} from the p-Pep/ Cu^{2+} hydrogels after different cycles. For b, d and f, values represent the mean and standard deviation ($n = 3$ independent experiments).

8. R^2 values need to be provided for the linearly regressed fatigue threshold limits. The figures make it very difficult to understand the goodness of fit. Also are these measurements in replicates? There is no standard deviation in the plots. Overall, it is difficult to assess confidence in these extrapolations and comparisons to various tough hydrogels.

Response: We thank the reviewer for the comments. Now we have provided the R^2 values for the linearly regressed fatigue threshold limits in the figure legend of Fig. 4f. The R^2 values of the linear fitting to estimate fatigue thresholds for PAM, p-Pep and p-Pep/ Cu^{2+} hydrogels were 0.99, 0.99 and 0.95, indicating excellent linearity coefficients. Moreover, these measurements were conducted in replicates following the reviewer's suggestion and the standard deviations are shown in the plots. The standard deviations for all the plots were less than 20%. Taking the R^2 values and the standard deviation into consideration, the fatigue threshold of the p-Pep/ Cu^{2+} hydrogels can be reliably reported as $424 \pm 22 \text{ J m}^{-2}$. Therefore, we are confident to conclude that the fatigue threshold for the p-Pep/ Cu^{2+} hydrogels is much higher than most of the other tough hydrogels. The R^2 values of fitting and standard deviation of the data are included in the revised manuscript. (See Fig. 4f and figure legends of Fig. 4f on Page 17-18 in the revised manuscript)

Revisions:

Fig. 4. Mechanical properties of hydrogels made of metal ion-clad picot fibres. **a**, Typical stress–strain curves under tension for PAM, p-Pep and p-Pep/Cu²⁺ hydrogels. **b**, Typical strain–stress curves of p-Pep/Cu²⁺ hydrogels under tension at various peptide concentrations in the precursor of hydrogels (0%, 2%, 4%, 6%, and 9% w/v). The inset corresponds to the summarized Young’s modulus and work of rupture (W_r) of the p-Pep/Cu²⁺ hydrogels. Values represent the mean and standard deviation ($n=3$ independent samples). **c**, Typical stretching–relaxation cycles of PAM, p-Pep and p-Pep/Cu²⁺ hydrogels at a strain of 4 mm mm⁻¹. **d**, Stretching–relaxation cycles for p-Pep/Cu²⁺ hydrogels subjected to various strains (4, 8, 12, 16, and 20 mm mm⁻¹). The curves are offset for clarity, and the overlapping curves are shown in the inset image. **e**, Stress–strain curves for 100 consecutive stretching–relaxation cycles without any stops for p-Pep and p-Pep/Cu²⁺ hydrogels. The cycle numbers are 1, 10, 20, 30, 40, 50, 60, 70, 80, 90 and 100. **f**, Extension of cracks per cycle as a function of the energy release rate for the PAM, p-Pep and p-Pep/Cu²⁺ hydrogels. The R^2 values of the linear fitting to estimate fatigue thresholds for PAM, p-Pep and p-Pep/Cu²⁺ hydrogels were 0.99, 0.99 and 0.95, respectively. The data points represent the mean and standard deviation ($n=3$ independent experiments). For **a**, **c**, **d**, **e**, and **f**, the peptide concentration in the hydrogel precursors was 6% w/v.

Reviewer #2 (Remarks to the Author):

The authors report the p-Pep/Cu²⁺ hydrogel having polyacrylamide skeletal network with polypeptide motifs. The polypeptides form supramolecular peptide fibers inside the gel, which can be further reinforced by Cu²⁺ ions. The obtained gels exhibit large stretchability, high toughness, and fast recovery.

This material belongs to tough soft materials based on the sacrificial bond principle. While typical materials based on this principle have two components, which are the brittle sacrificial component and stretchy matrix, the authors try to integrate these two roles into a single network as illustrated in Fig. 1. This idea is novel and interesting.

On the other hand, the reviewer does not think that the actual gel has the structure drawn in Fig. 1. Also, the reviewer is not sure how this work is essentially novel. The p-Pep/Cu²⁺ hydrogel can be classified as dual cross-linking gels with physical and chemical bonds. It has been generally reported that addition of physical bonding to a chemical gel improves stretchability and toughness (ex. K. Mayumi et al., *Ext. Mech. Lett.* 2016, 6, 52). The difference between the authors' gel and the reported dual cross-linking gels is not explained enough.

Overall, the novelty of the ideas and the mechanical properties make this manuscript worthy of publication in *Nature Communications*. On the other hand, the structure and mechanism of good mechanical properties should be reconsidered. The reviewer suggests major revision.

Response: We thank the reviewer for his/her comments that “the novelty of the ideas and the mechanical properties make this manuscript worthy of publication in *Nature Communications*”. We also understand that the structure of the hydrogel network should be further clarified to further highlight the novelty. We thank the reviewer for recommending the paper about using physical bonding to improve the stretchability and toughness of chemical gels (K. Mayumi et al., *Ext. Mech. Lett.* 2016, 6, 52) to our attention and we have cited this excellent paper in the revised manuscript (Reference No. 9). We have revised the manuscript according to the reviewers' valuable comments and hope the reviewer will find the revised manuscript is now acceptable in *Nature Communications*.

Comments:

1. The illustrations in fig. 1b and fig. S10 are unrealistic and should be extensively modified. While release of the PAAm hidden length after dissociation of the p-Pep/Cu²⁺ picot fibers possibly occurs in the gel, the gel should not have such ideal structure where PAM “loops” connect the adjacent peptides. According to the procedure, AM and polymerizable peptides were randomly co-polymerized to form the gel.

Regarding to this comment, the term “PAM loops” is not suitable. Instead of “loops”, “hidden length” can be used.

Response: We thank the reviewer for the comments. We agree with the reviewer that the peptide fibres should not have such ideal structure with PAM “loops” connecting the adjacent peptides. According the preparation procedure of p-Pep/Cu²⁺ hydrogels, the peptide self-assembled into fibres and then polymerized through the copolymerization with acrylamide. As shown the new version of Fig. 1b and Supplementary Fig. 10 (now has been changed into Supplementary Fig. 11), the picots formed by PAM are at different lengths and the peptides connected with each other randomly instead of only adjacent peptides. Besides, we used “hidden length” instead of “loop” in the revised manuscript, following the reviewer’s comments. The new comments as well as revised figures are included in the revised manuscript and Supplementary Information. (See the first paragraph on Page 6 and Fig. 1 on Page 7 in the revised manuscript; See Supplementary Fig. 11 on Page 15 in the revised Supplementary Information; See the changes of “loop” in the abstract on Page 2 and the second paragraph on page 5 in the revised manuscript)

On the other hand, we further confirmed that the acrylamide would zip into the hidden length of the picot peptide fibres using AFM imaging and AFM-based single-molecule force spectroscopy (SMFS). The metal ion-clad picot fibres formed by the copolymerization of GK₁₁ and acrylamide at different ratios of acrylamide and peptide were prepared and scanned using AFM. The diameters of the metal ion-clad picot fibres increased from 2.2 nm to 2.9 nm with the molar ratio of acrylamide and GK₁₁ increasing

from 16.3 to 32.6. This indicates that the polyacrylamide hidden length was indeed coated on the peptide fibres and longer hidden length led to thicker polyacrylamide layer on the peptide fibres.

Then the SMFS of metal ion-clad picot fibres was studied. As show in Supplementary Fig. 6a, the folded polymers were picked up from random positions along their contour by the cantilever and stretched to mechanically unfold the peptide structures and release the polyacrylamide hidden length. This gives rise to sawtooth-like traces (colored in blue) with each peak corresponding to the rupture of the interactions between a pair of peptide strands (Supplementary Fig. 6b and c). These peaks can be adequately fitted using worm-like chain (WLC) models (red lines). The contour length increment of each peak (ΔL) was considered as the hidden length between peptides. With the molar ratio of acrylamide and GK₁₁ increasing from 16.3 to 32.6, the average hidden length increased from 5.4 to 9.5 nm (Supplementary Fig. 6d and e). The measured values of the hidden length were close to the theoretical values of 4.9 nm and 9.8 nm, further revealing that most of the acrylamide were integrated to the hidden length of picot fibres. The new results and discussion were included in the revised manuscript and Supplementary Information. (See the second paragraph on Page 10 in the revised manuscript; See the section of “AFM based single molecule force spectroscopy (SMFS) measurements” on Page 4 and Supplementary Fig. 5-6 on Page 12-13 in the revised Supplementary Information)

Revisions:

...Driven by the self-assembly of the peptides, the links between peptides are zipped as picots to contribute to the hidden length (Fig. 1b). The picots are at different lengths and the peptides connected with each other randomly...

...Here we present a type of hydrogels comprising hierarchical structures of picot fibres made of copper-bound self-assembling peptide strands with zipped flexible hidden length. The redundant hidden lengths allow the fibres to be extended to dissipate

mechanical load without reducing network connectivity, making the hydrogels robust against damage...

...Inspired by the structures of biological networks, we propose a new type of hierarchical structure made of picot fibres (p-fibres) consisting of self-assembled metal ion-clad peptide β -strands interconnected by flexible hidden lengths...

Furthermore, the hidden length in picot fibres can be tuned by adjusting the acrylamide concentration. As shown in Supplementary Fig. 5, the metal ion-clad picot fibres became thicker at the higher ratio of acrylamide and peptide, indicating that increasing ratio of polyacrylamide led to the longer hidden length. This was also confirmed by the AFM based single molecule force spectroscopy (SMFS) of metal ion-clad picot fibres (Supplementary Fig. 6). The hidden length increased from ~5.4 nm to ~9.5 nm when the molar ratio of acrylamide and ACLT-GK₁₁ increased from 16.3 to 32.6. The measured values of the hidden length were close to theoretical values of 4.9 nm and 9.8 nm, further revealing that most of the acrylamide was integrated to the hidden length of picot fibres.

AFM based single molecule force spectroscopy (SMFS) measurements

For the preparation of cantilevers used in SMFS, silicon nitride (Si₃N₄) cantilevers (MLCT, Bruker, USA) were first cleaned with Milli-Q water, and then placed in a chromic mixture (chromic acid) at 80 °C for 30 min. After that, the cantilevers were washed with deionized water, then ethanol, and dried under a steam of nitrogen. For the preparation of metal ion-clad picot fibres absorbed on glass substrates used in SMFS, ACLT-GK₁₁ (43.2 mM) and acrylamide (703 mM or 1406 mM) were copolymerized in deionized water and diluted for 3 times using 1 M Tris buffer (pH=7.60, containing 300

mM KCl and 150 mM CuCl₂) to induce self-assembly and ion binding. For the preparation of substrates, the glass substrates were cut into 1 × 1 cm² slides and soaked in a freshly prepared chromic mixture overnight. Then the substrates were washed with deionized water and ethanol, and then dried under a stream of nitrogen. The as prepared ion-clad picot fibre solution (50 μL) was dropped on the glass substrate and allowed to absorb for 30 min. The solution was removed, and the substrate was washed using Tris buffer for 3 times to remove the unabsorbed fibres.

AFM force spectroscopy experiments were carried out on a commercial AFM (JPK Nanowizard II). The force–distance curves were recorded by commercial software from JPK and analyzed by custom-written procedures in Igor pro 6.37 (Wavemetrics, Inc.). All the experiments were conducted at the room temperature (22 °C) and performed in 1 M Tris buffer (pH=7.60, containing 300 mM KCl). Soft silicon nitride MLCT-D cantilevers with typical spring constants of 30–45 pN nm⁻¹ were used for all experiments and calibrated using the thermal tune method. Typically, the cantilever was brought in contact with the substrate and held at the surface for 3 s, then retracted at a constant velocity of 2.0 μm s⁻¹. The sampling rate was 8 kHz. The contour length increment was determined by fitting the force peaks using the worm-like chain model with persistence lengths in a range of ~0.2–0.4 nm. The contour length increment of each peak (ΔL) was considered as the hidden length between peptides.

Fig. 1. Hydrogel design based on metal ion-clad picot fibres. **a**, Schematic of a conventional double-network hydrogel. The physically crosslinked network is designed to dissipate energy, and the covalently crosslinked network is used to provide hidden length. **b**, Schematic of the hydrogel constructed by picot fibres made of self-assembling peptide strands with zipped flexible hidden lengths. Upon deformation of the hydrogel, the picot fibres are extended to dissipate energy efficiently and release the polyacrylamide hidden length without reducing the network connectivity of the hydrogels. **c**, The peptide sequence and synthetic scheme for the metal ion-clad picot fibres. **d**, The self-assembled structure of GK₁₁ (left) and a magnified region of this diagram showing the binding position of Cu²⁺ (right). The hydrogen bonds are shown as grey dashed lines. Histidine is coloured in green in the ribbon representation in the

left image and represented by sticks to illustrate the coordination bonds in the right image.

Supplementary Figure 5. AFM imaging of metal ion-clad picot fibres (pGK₁₁/Cu²⁺) at different acrylamide:ACLT-GK₁₁ ratios. a, AFM image (left) and height profile (right) of metal ion-clad picot fibres (pGK₁₁/Cu²⁺) at the acrylamide:ACLT-GK₁₁ ratio of 16.3. The height profile at right corresponds to the cross line (green) in the left image. Scale bar = 1 μ m. b, Diameter distribution of metal ion-clad picot fibres (pGK₁₁/Cu²⁺) at the acrylamide:ACLT-GK₁₁ ratio of 16.3. c, AFM image (left) and height profile (right) of metal ion-clad picot fibres (pGK₁₁/Cu²⁺) at the acrylamide:ACLT-GK₁₁ ratio of 32.6. The height profile at right corresponds to the cross line (green) in the left image. Scale bar = 1 μ m. d, Diameter distribution of metal ion-clad picot fibres (pGK₁₁/Cu²⁺) at the acrylamide:ACLT-GK₁₁ ratio of 32.6.

Supplementary Figure 6. SMFS of metal ion-clad picot fibres at different ratios of acrylamide and ACLT-GK₁₁. a, Schematic of the AFM-based SMFS for metal ion-clad picot fibres. b, c, Representative force–distance curves for the rupture of the metal ion-clad picot fibre at acrylamide:ACLT-GK₁₁ ratios of 16.3 (b) and 32.6 (c). The pulling speed was $2 \mu\text{m s}^{-1}$ and each peak corresponds to an individual rupture event between a pair of peptide strands. Red lines correspond to worm-like chain (WLC) fitting to the rupture events using the persistence length of 0.2-0.4 nm. ΔL indicates the hidden length between fractured peptides. d, e, Histograms of hidden lengths in metal ion-clad picot fibres at the acrylamide:ACLT-GK₁₁ ratio of 16.3 (d) and 32.6 (e). The measured hidden lengths (ΔL) were 5.4 nm (N=230) and 9.5 nm (N=316), respectively. Red lines correspond to a Gaussian fit.

Supplementary Figure 11. Schematic of the step-by-step release of the hidden length upon rupture of neighbouring β strands in the picot fibres.

2. According to the proposed structure, mechanical properties of the p-Pep/Cu²⁺ hydrogel should be dominated by enthalpic contribution of inter-peptide hydrogen bonding, not by entropic contribution. Thus, analysis of the elasticity based on the classical rubber elasticity is not suitable.

Response: We thank the reviewer for the comments. The reviewer is correct that “the mechanical properties of the p-Pep/Cu²⁺ hydrogel should be dominated by enthalpic contribution of inter-peptide hydrogen bonding, not by entropic contribution”. At low strains, the peptide fibres remain integrated without unfolding. Therefore, it is possible to estimate the elasticity of the hydrogels at this region based on rubber elasticity and considering the peptide fibres as unextendible rods. Indeed, the calculations based on rubber elasticity matched with our experimentally obtained values. At high strains, the elasticity of the hydrogels can be affected by the rupture of the peptide fibres as well as the alignment of the fibres to the force direction, which is quite complicated. Note that, the unfolding of the peptide fibres contributes largely to the fracture strain and toughness of the hydrogels. We have clarified this in the revised manuscript. (See the middle of Page 16 and the first paragraph on Page 17 in the revised manuscript)

Revisions:

...At low strains, the peptide fibres remain integrated without unfolding. Thus, the polyacrymide remains as the hidden length and does not contribute to the elasticity of the p-Pep/Cu²⁺ hydrogels. The hydrogels are formed by 4-armed PEG-ACLT with the picot fibres as unextendible crosslinkers. The elasticity of the hydrogels can be predicted using the classical rubber elasticity theory...

...It is worth noting that at high strains, the elasticity of the hydrogels can be affected by the rupture of the peptide fibres as well as the alignment of the fibres to the force direction, which is quite complicated. Moreover, the unfolding of the peptide fibres contributes largely to the fracture strain and toughness of the hydrogels.

3. As mentioned before, basically the p-Pep/Cu²⁺ hydrogel is a kind of dual cross-linking gels with physical and chemical bonds. In comparison with other gels having physical bonding, what is the key physical factor to obtain the good mechanical properties including fast recovery? The reviewer requests the authors to add more comments about this.

Response: We thank the reviewer for the comments. For hydrogels with normal network structures, stiffness is linearly proportional to the crosslinking density, and toughness is inversely proportional to the square root of the crosslinking density of hydrogels. Therefore, both chemical and physical crosslinks stiffen hydrogels with the trade-off of reducing their toughness. In contrast, in the p-Pep/Cu²⁺ hydrogels, the physical bonds were mainly used to build the metal ion-clad picot fibres. Each fibre can be considered as an extensible crosslinker. Increasing physical interactions by raising the peptide or metal ion concentrations only increases the total available sacrificial bonds in the fibres without affecting the crosslinking density of the hydrogels, thus resolving the stiffness-toughness trade-off common in polymer networks.

Besides, the unique hydrogel network structure formed by crosslinked picot fibres (Fig. 1b) also contributed to the fast mechanical recovery of the p-Pep/Cu²⁺ hydrogels. Rupture and reformation of the picot fibres mainly involved the recovery of

intramolecular interactions which was a localized process independent of the recovery for the overall hydrogel network. In contrast, in typical hydrogels with dual physical and covalent crosslinks, the randomly distributed physical crosslinks are prone to rebinding at incorrect positions. Fully recovering the mechanical properties of the hydrogels requires the break and reform of the misregistered physical crosslinks, thus slowing the recovery of hydrogels. Moreover, most Cu^{2+} remained attached to the peptides even the fibres were unfolded upon stretching the hydrogels. These Cu^{2+} ions may serve as the nuclei for the folding of picot fibres, as the fibres without Cu^{2+} showed reduced recovery rates. See the Response to the Q7 of Reviewer #1 for details. The new data and discussion have been included in the revised manuscript and Supplementary Information. (See the end of Page 14, the first paragraph on Page 15 and the end of Page 19 in the revised manuscript; See Supplementary Fig. 22 on Page 24 in the revised Supplementary Information)

Revisions:

...For hydrogels with normal network structures, stiffness is linearly proportional to the crosslinking density⁵¹, and toughness is inversely proportional to the square root of the crosslinking density.^{52, 53} Therefore, both chemical and physical crosslinks stiffen hydrogels with the trade-off of reducing their toughness. In contrast, in the p-Pep/ Cu^{2+} hydrogels, the physical bonds were mainly used to build the metal ion-clad picot fibres. Each fibre can be considered as an extensible crosslinker. Increasing physical interactions by raising the peptide or metal ion concentrations only increases the total available sacrificial bonds in the fibres without affecting the crosslinking density of the hydrogels, thus resolving the stiffness-toughness trade-off common in polymer networks...

...Moreover, the p-Pep/ Cu^{2+} hydrogels recovered even faster than the p-Pep hydrogels. It is probably because Cu^{2+} remained attached to the unfolded picot fibres and thus facilitated reformation of the picot fibres in the unloaded hydrogels, as indicated by the almost unchanged mechanical strength and negligible Cu^{2+} release of p-Pep/ Cu^{2+}

hydrogels under stretching-relaxation cycles in Tris buffer (Supplementary Fig. 22). These Cu^{2+} ions may serve as the nuclei for the folding of picot fibres, as the fibres without Cu^{2+} showed reduced recovery rates.

Supplementary Figure 22. Mechanical property and Cu^{2+} release of p-Pep/ Cu^{2+} hydrogels under stretching-relaxation cycles in Tris buffer (pH=7.60, containing 300 mM KCl). a, Typical stress–strain curves for p-Pep and p-Pep/ Cu^{2+} hydrogels under stretching-relaxation cycles. The cycle numbers are 1, 10, 20, 40, 60, 80 and 100. b, Summary of the maximum stress and energy dissipation for p-Pep/ Cu^{2+} hydrogels under stretching-relaxation cycles. c, UV-vis spectra for mixture of sodium diethyldithiocarbamate (DDTC-Na) and Cu^{2+} at different concentrations of Cu^{2+} (1-50 μM). The concentration of DDTC-Na was 250 μM . d, Calibration curve of Cu^{2+} concentrations and OD_{452nm} (UV-vis absorbance at 452 nm). e, UV-vis spectra of the hydrogel leachates after different cycles in the presence of DDTC-Na. f, Accumulated

percentage of released Cu^{2+} from the p-Pep/ Cu^{2+} hydrogels after different cycles. For b, d and f, values represent the mean and standard deviation ($n=3$ independent experiments).

4. For fig. S16, it is not clear how the authors estimate the theoretical value of 2.0. Please show the estimation process. Is it calculated based on the swelling ratio difference?

Response: We thank the reviewer for the comments. The calculation of the theoretical values of the average length between two neighbouring cross-linking points after and before the release of the hidden length was shown in Supplementary Fig. 16 (now has been changed into Supplementary Fig. 18). Considering the network of the p-Pep/ Cu^{2+} hydrogel, the average length between two neighbouring cross-linking points before the release of the hidden length can be determined as Eq. (2).

$$L_1 = L_{\text{acrylamide+pep}} + 2R_{\text{PEG}} \quad (2)$$

where $L_{\text{acrylamide+pep}}$ is the length of the picot fibre between two neighbouring cross-linking points and R_{PEG} is the mean square end-to-end distance of a linear PEG (5 kDa). $L_{\text{acrylamide+pep}}$ can be determined as $\frac{C_{\text{pep}} \times (W_{\text{pep}} + L_{\text{gap}})}{4C_{\text{PEG}}}$, in which C_{pep} is the molar concentration of ACLT-GK₁₁ peptide, W_{pep} is the average width of an ACLT-GK₁₁ peptide (width of a β sheet), L_{gap} is the average length of the gap between adjacent peptide in the assembled GK₁₁, and C_{PEG} is the molar concentration of 4-armed PEG-ACLT. Following the worm-like chain (WLC) model, R_{PEG} is determined as $R_{\text{PEG}} = \sqrt{2\zeta_{\text{PEG}}L_{\text{PEG}}}$, in which ζ_{PEG} and L_{PEG} are the persistence length and the contour length of PEG, respectively. We used the ζ_{PEG} of 0.38 nm and L_{PEG} of 40 nm according to the previous literatures. The average width of a peptide (width of the β sheet) was 0.35 nm and the gap between adjacent peptide was 0.30 nm according to the simulation.

The average length between two neighbouring cross-linking points after the release of the hidden length can be determined as Eq. (3).

$$L_2 = R_{\text{acrylamide+pep}} + 2R_{\text{PEG}} \quad (3)$$

in which $R_{\text{acrylamide+pep}}$ is the mean square end-to-end distance of copolymerized acrylamide and peptide, R_{PEG} is the mean square end-to-end distance of PEG. The mean square end-to-end distance of copolymerized acrylamide and peptide can be determined following the WLC model.

$$R_{\text{acrylamide+pep}} = \sqrt{2 \zeta_{\text{acrylamide+pep}} L_{\text{acrylamide+pep}}} \quad (4)$$

where $\zeta_{\text{acrylamide+pep}}$ and $L_{\text{acrylamide+pep}}$ are the persistence length and contour length of copolymerized acrylamide and peptide, respectively. The persistence length of the copolymer of acrylamide and peptide was taken as 0.4 nm. The contour length of copolymerized acrylamide and peptide can be determined as Eq. (5).

$$L_{\text{acrylamide+pep}} = \frac{C_{\text{acrylamide}} + C_{\text{pep}}}{4C_{\text{PEG}}} \times 2L_{\text{C-C}} \quad (5)$$

where $L_{\text{C-C}}$ is the length of the C-C bond, $C_{\text{acrylamide}}$, C_{pep} and C_{PEG} are the molar concentrations of acrylamide, ACLT-GK₁₁ and PEG. The length of C-C bond was considered as 0.15 nm. Finally, the theoretical ratio of the average length between two neighbouring cross-linking points after and before the release of the hidden length was calculated as $\frac{L_2}{L_1} = 2.0$. The details about the estimation of the theoretical value of 2.0 is included in the section of ‘‘Calculation of the hidden length in the hydrogel network’’ in the Supplementary Information. (See the section of ‘‘Calculation of the hidden length in the hydrogel network’’ on Page 7-9 in the revised Supplementary Information)

Revisions:

Calculation of the hidden length in the hydrogel network

Considering the network of the p-Pep/Cu²⁺ hydrogel, the average length between two neighbouring cross-linking points before the release of the hidden length (top of Supplementary Fig. 18) can be determined as Eq. (2).

$$L_1 = L_{\text{acrylamide+pep}} + 2R_{\text{PEG}} \quad (2)$$

where $L_{\text{acrylamide+pep}}$ is the length of the picot fibre between two neighbouring cross-

linking points and R_{PEG} is the mean square end-to-end distance of a linear PEG (5 kDa). $L_{acrylamide+pep}$ can be determined as $\frac{C_{pep} \times (W_{pep} + L_{gap})}{4C_{PEG}}$, in which C_{pep} is the molar concentration of ACLT-GK₁₁ peptide, W_{pep} is the average width of an ACLT-GK₁₁ peptide (width of a β sheet), L_{gap} is the average length of the gap between adjacent peptide in the assembled GK₁₁, and C_{PEG} is the molar concentration of 4-armed PEG-ACLT. Following the worm-like chain (WLC) model, R_{PEG} is determined as $R_{PEG} = \sqrt{2\zeta_{PEG}L_{PEG}}$, in which ζ_{PEG} and L_{PEG} are the persistence length and the contour length of PEG, respectively. We used the ζ_{PEG} of 0.38 nm and L_{PEG} of 40 nm according to the previous literatures.¹⁵⁻¹⁷ The average width of a peptide (width of the β sheet) was set as 0.35 nm and the gap between adjacent peptide was 0.30 nm according to the simulation.

The average length between two neighbouring cross-linking points after the release of the hidden length (bottom of Supplementary Fig. 18) can be determined as Eq. (3).

$$L_2 = R_{acrylamide+pep} + 2R_{PEG} \quad (3)$$

in which $R_{acrylamide+pep}$ is the mean square end-to-end distance of copolymerized acrylamide and peptide, R_{PEG} is the mean square end-to-end distance of PEG. The mean square end-to-end distance of copolymerized acrylamide and peptide can be determined following the WLC model.

$$R_{acrylamide+pep} = \sqrt{2\zeta_{acrylamide+pep}L_{acrylamide+pep}} \quad (4)$$

where $\zeta_{acrylamide+pep}$ and $L_{acrylamide+pep}$ are the persistence length and contour length of copolymerized acrylamide and peptide, respectively. 0.4 nm was used as the persistence length of copolymerized acrylamide and peptide.¹⁸ The contour length of copolymerized acrylamide and peptide can be determined as Eq. (5).

$$L_{acrylamide+pep} = \frac{C_{acrylamide} + C_{pep}}{4C_{PEG}} \times 2L_{C-C} \quad (5)$$

where L_{C-C} is the length of the C-C bond, $C_{acrylamide}$, C_{pep} and C_{PEG} are the molar concentrations of acrylamide, ACLT-GK₁₁ and PEG. The length of C-C bond was considered as 0.15 nm. Finally, the theoretical ratio of the average length between two

neighbouring cross-linking points after and before the release of the hidden length was calculated as $\frac{L_2}{L_1} = 2.0$.

Based on the classical rubber elasticity theory and Gaussian chain statistics¹⁹⁻²², the stress of the hydrogel can be described as Eq. (6).

$$\sigma = G \left(\lambda - \frac{1}{\lambda^2} \right) = \frac{\rho RT}{M_c} \left(\lambda - \frac{1}{\lambda^2} \right) \quad (6)$$

in which σ is the stress, $G = \frac{\rho RT}{M_c}$ is the shear modulus, ρ is the density of the polymer in the network, M_c is the molecular weight between two neighbouring cross-linking points, R is the gas constant, and λ is the extension ratio. Young's modulus E is defined as $E = \frac{\partial \sigma}{\partial \lambda}$ and $E = 3G$ when $\lambda = 1$. The ratio of Young's modulus for the same hydrogel before and after the release of the hidden length is $\varepsilon = \frac{\rho_1 M_{c2}}{\rho_2 M_{c1}}$. According to the Young's modulus and swelling ratio of the hydrogel before and after the treatment of GuHCl in experiments, the ratio of M_c after and before the release of the hidden length is $\sim 1.8 \pm 0.3$. We assume that the length of the polymer chain in the hydrogel network is directly proportional to its molecular weight. As a result, the ratio of the average length between two neighbouring cross-linking points after and before the release of the hidden length in experiments was calculated as $\sim 1.8 \pm 0.3$.

5. "As these sacrificial bonds cannot reform efficiently, the hydrogels lack a mechanism to prevent crack propagation and are susceptible to fatigue under cyclic loading"; the reviewer is unsure of the relationship between them. Is the slow reformation of sacrificial bonds really the reason of the poor fatigue resistance?

Response: We thank the reviewer for the comments. Studies by Lake and coworkers on rubbers (*Rubber Chem. Technol.* 1995, 68, 435–460; *Proc. R. Soc. Lond. A Math. Phys. Sci.* 1967, 300, 108–119) and Suo and coworkers on hydrogels (*Eur. J. Mech. A, Solids* 2019, 74, 337–370, *ACS Macro Lett.* 2018, 7, 312–317, *ACS Macro Lett.* 2019, 8, 17–23) suggested fatigue threshold is determined by the covalent network, whereas the noncovalent/reversible interactions make negligible contributions. However, this role cannot quantitatively match the fatigue threshold of soft materials containing

dynamic bonds. Recently, Gong et. al. studied the role of dynamic bonds on fatigue threshold of tough hydrogels (*Proc. Natl. Acad. Sci. USA* 2022, 119 (20), e2200678119.). They raised the point that in the viscoelastic regime, the dynamic bonds carry load, protecting the permanent bonds from breaking, and thereby contribute to the fatigue threshold. The fatigue resistance of soft materials can be improved by designing dynamic bonds with a relatively long relaxation time so that the observation (working) time window falls in the viscoelastic regime. As a result, the slow reformation of sacrificial bonds after fracture cannot provide long-term protections on the permanent bonds in cyclic loading, thus leading to the poor fatigue resistance. Moreover, the short relaxation time of sacrificial bonds with slow reformation rates prevent the observation (working) time window falling in the viscoelastic regime, thus decreasing the fatigue resistance. Now we have included the new comments and cited the new reference in the revised manuscript. (See the bottom of Page 3, the end of Page 20, the first paragraph on Page 21 and reference No. 30 in the revised manuscript)

Revisions:

...As these sacrificial bonds cannot reform efficiently, the hydrogels lack a mechanism to prevent crack propagation and are susceptible to fatigue under cyclic loading³⁰...

...A recent study showed that dynamic interactions can only carry load and contribute to the fatigue resistance when their lifetimes are longer than the timescale of the periodic mechanical load.³⁰ Note that both the lifetimes and recovery rates of typical physical interactions are considerably reduced by applied force.^{64, 65} Therefore, they cannot carry load under cyclic stretch to prevent crack propagation. We attribute the outstanding fatigue resistance of the p-Pep/Cu²⁺ hydrogel to the high mechanical stability and fast reassembly of the metal ion-clad picot fibres. In addition, the release of hidden length can also prevent crack propagation and increase the fatigue threshold according to the Lake-Thomas model^{52, 66}...

6. “it remains challenging to achieve high strength, high toughness, robust fatigue

resistance, and rapid recovery in the same synthetic hydrogel”; please check the two recent papers published in Science, Liu et al., Science 2021, 372, 1078 / Kim et al., Science 2021, 374, 212. These works have realized high toughness of elastic hydrogels (with rapid recovery) with different strategies.

Response: We thank the reviewer for the comments. We agree with the reviewer that Liu et al. and Kim et al. presented different strategies to realize high toughness of elastic hydrogels (with rapid recovery). Liu et al. used the damageless reinforcement strategy of strain-induced crystallization to build high toughness and rapid recovery. Kim et al. used dense entanglements which greatly outnumber cross-links in hydrogels to realize high toughness, strength, and fatigue resistance. However, they did not combine all the high strength, high toughness, robust fatigue resistance, and rapid recovery in the same synthetic hydrogel, which can be addressed to a certain extent using the design of picot fibres in our work. We have cited these works and added comments in the revised manuscript. (See the end of Page 3, the first paragraph on Page 4 and reference No. 31-32 in the revised manuscript)

Revisions:

...Recently, some efforts have partially overcome the trade-offs among these mechanical properties using strain-induced self-reinforcement³¹, chain entanglement³² or mechanochemically activable hidden length³³. For example, Liu et al. used the damageless reinforcement strategy of strain-induced crystallization to build high toughness and rapid recovery.³¹ Kim et al. used dense entanglements which greatly outnumber cross-links in hydrogels to realize high toughness, strength, and fatigue resistance.³² Moreover, the hidden length in fibril structures or folded proteins can provide extra extensibility after being released, thus enhancing the toughness of hydrogels.³³⁻³⁶ ...

7. The p-Pep/Cu²⁺ hydrogel is more deformable than the PAM gel. While the authors assume that this is due to formation of the peptide fibers, it is also possible that lower swelling ratio of the p-Pep/Cu²⁺ gel ensures its large deformability because swelling of

a gel generally decreases its deformability.

Response: We thank the reviewer for the comments. We agree with the reviewer that the lower swelling ratio of the p-Pep/Cu²⁺ hydrogel would lead to the large deformability. Actually, we think that the large deformability due to lower swelling ratio is consistent with the formation of the peptide fibres. Since GK₁₁ peptide can form fibres in hydrogels, the acrylamide was zipped as the hidden length of the peptide fibres. Hence, acrylamide would not contribute to the swelling of hydrogels before the release of hidden length, leading to the low swelling ratio of p-Pep/Cu²⁺ hydrogels. During the deformation, fracturing the peptide fibres can release the hidden length without reducing network connectivity of the hydrogels, leading to the significantly enhanced deformability. This is further confirmed by the increased fracture strain of hydrogels at higher acrylamide concentrations (Supplementary Fig. 15c-d). Now, we have included the discussion in the revised manuscript. (See the second paragraph on Page 12 in the revised manuscript and Supplementary Fig. 15 on Page 17 in the revised Supplementary Information)

Revisions:

...The deformability of p-Pep/Cu²⁺ hydrogels can be mainly attributed to the zipped acrylamide in hidden length of the peptide fibres, which also contributed to the low swelling ratio. The released hidden length of peptide fibres during deformations led to the significantly enhanced deformability...

Supplementary Figure 15. Mechanical properties of the p-Pep/Cu²⁺ hydrogels with various 4-armed PEG-ACLT or acrylamide concentrations. a, Uniaxial stretching stress–strain curves of p-Pep/Cu²⁺ hydrogels with different 4-armed PEG-ACLT concentrations (0%, 1%, 3%, 5% and 7% w/v) in the precursors during preparation. b, Summaries of Young’s modulus and **work of rupture** of p-Pep/Cu²⁺ hydrogels at different 4-armed PEG-ACLT concentrations (0%, 1%, 3%, 5% and 7% w/v) in the precursors during preparation. c, Uniaxial stretching stress–strain curves of p-Pep/Cu²⁺ hydrogels at different acrylamide concentrations in the precursors during preparation (225, 325 and 450 mg mL⁻¹). d, Summaries of Young’s modulus and **work of rupture** of p-Pep/Cu²⁺ hydrogels at different acrylamide concentrations in the precursors during preparation (225, 325 and 450 mg mL⁻¹). For b and d, values represent the mean and standard deviation (n = 3 independent samples).

8. About fig. S8, use of SEM to estimate mesh size of a wet gel is generally not suitable. Such a “mesh” in a dried gel is probably an artifact and formed during a drying process.

Response: We thank the reviewer for the comments. We totally agree with the reviewer that SEM cannot represent the precise mesh size of a wet hydrogel quantitatively. SEM images can only be used to distinguish the mesh size of different hydrogels qualitatively, because the original mesh structures cannot be fully maintained in the absence of the solvent after lyophilization. To avoid confusion, now we have removed the SEM images

and the statement of mesh size in the revised manuscript. (See the end of Page 11 in the revised manuscript)

Revisions:

~~... Scanning electron microscopy (SEM) confirmed that the picot fibre hydrogels had smaller mesh sizes than the PAM hydrogels and that the mesh size further decreased with ion coordination (Supplementary Fig. 8). All these results indicate the hydrogel network structures are affected by the formation of picot fibres and metal ion binding.~~

9. Please define the term “toughness” carefully. The authors used the term “toughness” for two different physical quantities. One is the area below the tension stress-strain curve until fracture with the unit of J m^{-3} . The other is the fracture energy with the unit of J m^{-2} , which is called “toughness” in the introduction and conclusion. While the latter should be physically called as toughness, in this field both values are often called as toughness. The reviewer suggests using a different name for the former instead of “toughness” to avoid confusion.

Regarding to this, the authors wrote “toughness is inversely proportional to the square root of the crosslinking density”. This “toughness” is the latter one, not the former one.

Response: We thank the reviewer for the insightful comments. We agree with the reviewer that the term “toughness” should be defined clearly in the manuscript. Following the reviewer’s suggestion, the area below the tension stress-strain curve until fracture with the unit of J m^{-3} is defined as the work of rupture (W_r) (*Liu et al., Science 2021, 372, 1078*). The fracture energy with the unit of J m^{-2} is defined as toughness. The terms have been revised throughout the manuscript and Supplementary Information. (See the end of Page 13, the second paragraph on Page 14, the second paragraph on Page 15, the second paragraph on Page 25 and Fig. 4 on Page 17 in the revised manuscript; See Supplementary Fig. 15-16 on Page 17-18 and Supplementary Table 1-2 on Page 25-26 in the revised Supplementary Information)

Revisions:

...The fracture stress, Young's modulus and the work of rupture (\$W_r\$ ) for the p-Pep/Cu²⁺ hydrogel were significantly higher than those of the PAM and p-Pep hydrogels (Fig. 4a and Supplementary Table 1)...

...The work of rupture of PAM, p-Pep and p-Pep/Cu²⁺ hydrogels were ~0.4, 7.3 and 38.1 MJ m⁻³, respectively...

...The fracture stress and work of rupture also increased at increasing GK₁₁ concentrations at low peptide concentrations up to 6% and dropped when further increasing the peptide, presumably due to the aggregation of the peptide fibres (Supplementary Fig. 12)...

...The fracture strain and work of rupture of the p-Pep/Cu²⁺ hydrogels reached more than 2500% and 50 MJ m⁻³ by simply increasing the concentration of acrylamide (Supplementary Fig. 15c-d). Work of rupture and strength of hydrogels were also enhanced by the picot fibres in the compressive mechanical tests (Supplementary Fig. 16)...

...The work of rupture (\$W_r\$ ) was calculated from the area below the tension stress-strain curve until fracture...

Fig. 4. Mechanical properties of hydrogels made of metal ion-clad picot fibres. **a**, Typical stress–strain curves under tension for PAM, p-Pep and p-Pep/Cu²⁺ hydrogels. **b**, Typical strain–stress curves of p-Pep/Cu²⁺ hydrogels under tension at various peptide concentrations in the precursor of hydrogels (0%, 2%, 4%, 6%, and 9% w/v). The inset corresponds to the summarized Young’s modulus and *work of rupture* (W_r) of the p-Pep/Cu²⁺ hydrogels. Values represent the mean and standard deviation ($n=3$ independent samples). **c**, Typical stretching–relaxation cycles of PAM, p-Pep and p-Pep/Cu²⁺ hydrogels at a strain of 4 mm mm⁻¹. **d**, Stretching–relaxation cycles for p-Pep/Cu²⁺ hydrogels subjected to various strains (4, 8, 12, 16, and 20 mm mm⁻¹). The curves are offset for clarity, and the overlapping curves are shown in the inset image. **e**, Stress–strain curves for 100 consecutive stretching–relaxation cycles without any stops for p-Pep and p-Pep/Cu²⁺ hydrogels. The cycle numbers are 1, 10, 20, 30, 40, 50, 60, 70, 80, 90 and 100. **f**, Extension of cracks per cycle as a function of the energy release rate for the PAM, p-Pep and p-Pep/Cu²⁺ hydrogels. The R^2 values of the linear fitting to estimate fatigue thresholds for PAM, p-Pep and p-Pep/Cu²⁺ hydrogels were 0.99, 0.99 and 0.95, respectively. Values represent the mean and standard deviation ($n=3$ independent experiments). For **a**, **c**, **d**, **e**, and **f**, the peptide concentration in the hydrogel precursors was 6% w/v.

Supplementary Figure 15. Mechanical properties of the p-Pep/Cu²⁺ hydrogels with various 4-armed PEG-ACLT or acrylamide concentrations. a, Uniaxial stretching stress–strain curves of p-Pep/Cu²⁺ hydrogels with different 4-armed PEG-ACLT concentrations (0%, 1%, 3%, 5% and 7% w/v) in the precursors during preparation. b, Summaries of Young’s modulus and **work of rupture** of p-Pep/Cu²⁺ hydrogels at different 4-armed PEG-ACLT concentrations (0%, 1%, 3%, 5% and 7% w/v) in the precursors during preparation. c, Uniaxial stretching stress–strain curves of p-Pep/Cu²⁺ hydrogels at different acrylamide concentrations in the precursors during preparation (225, 325 and 450 mg mL⁻¹). d, Summaries of Young’s modulus and **work of rupture** of p-Pep/Cu²⁺ hydrogels at different acrylamide concentrations in the precursors during preparation (225, 325 and 450 mg mL⁻¹). For b and d, values represent the mean and standard deviation (n = 3 independent samples).

Supplementary Figure 16. Mechanical properties of p-Pep/Cu²⁺ hydrogels under compression. a, Typical stress–strain curves under compression for PAM, p-Pep and p-Pep/Cu²⁺ hydrogels. b, Summaries of fracture strain and stress of the PAM, p-Pep and p-Pep/Cu²⁺ hydrogels under compression. c, Summaries of Young’s modulus and **work of rupture** of the PAM, p-Pep and p-Pep/Cu²⁺ hydrogels under compression. d, Typical stress–strain curves of p-Pep/Cu²⁺ hydrogels at various peptide concentrations in the precursor of hydrogels (0%, 2%, 4%, and 6% w/v) under compression. e, Summaries of fracture strain and stress of p-Pep/Cu²⁺ hydrogels at various peptide concentrations in the precursor of hydrogels (0%, 2%, 4%, and 6% w/v) under compression. f, Summaries of Young’s modulus and **work of rupture** of p-Pep/Cu²⁺ hydrogels at various peptide concentrations in the precursor of hydrogels (0%, 2%, 4%, and 6% w/v) under compression. For b, c, e, and f, values represent the mean and standard deviation (n = 3 independent samples).

Supplementary Table 1. Tensile mechanical properties of PAM, p-Pep and p-Pep/Cu²⁺ hydrogels containing different concentrations of peptides. Values represent the mean and standard deviation.

Peptide concentration (v/w)	Fracture strain (mm mm ⁻¹)	Fracture stress (MPa)	Young’s modulus (kPa)	Work of rupture (MJ m ⁻³)	Toughness (kJ m ⁻²)
PAM /	5.4 ± 0.8	0.18 ± 0.03	13.2 ± 3.2	0.4 ± 0.04	0.10 ± 0.01

p-Pep	6%	14.1 ± 0.9	1.35 ± 0.12	25.4 ± 5.9	7.3 ± 1.6	2.6 ± 0.6
	2%	7.6 ± 0.3	0.40 ± 0.03	20.9 ± 3.7	1.3 ± 0.4	--
p-Pep/Cu ²⁺	4%	16.8 ± 0.7	1.92 ± 0.17	29.5 ± 2.8	12.6 ± 1.6	--
	6%	22.5 ± 1.4	4.12 ± 0.37	71.5 ± 4.6	38.1 ± 3.6	25.3 ± 1.9
	9%	18.2 ± 1.2	3.66 ± 0.21	121.5 ± 14.0	29.1 ± 3.5	--

Supplementary Table 2. Mechanical properties of the p-Pep/Cu²⁺ hydrogel, and other tough and rapid recovery hydrogels from the literature.

Classification	Sample code	Water content (wt%)	Young's modulus (MPa)	Fracture strain (mm/mm)	Fracture stress (MPa)	Work of rupture (MJ m ⁻³)	Toughness (kJ m ⁻³)	Recovery time (min)	Recovery efficiency (%)	Ref. No.
Single-network hydrogel	p-Pep/Cu ²⁺	87	0.07	22.5	4.12	38.1	25.3	0	~100	This work
		82 (*)	0.1 (*)	25.0 (*)	5.47 (*)	55.7 (*)	--	--	--	
	Highly entangled hydrogel	~70	0.05	~4.5	0.4	--	2.2	--	--	23
	Polyampholytes gel	50-70	0.01-8	1.5-15	0.1-2	0.1-7.0	1.0-4.0	120	~100	24
	(FL) ₈ gel	70	0.016	4.5	0.035	--	--	20	~85	25
	DMAA-MAAc hydrogel	28	8	8	2	--	9.3	3 (37 °C)	~100	26
Dual-crosslinked hydrogel	P(urea-IL ₆ -SPMA ₆)-3d gel	~50	1.97	4.78	1.90	6.70	--	120	~85	27
	PIC gel	55	5.4	7.5	3.8	18.8	10.0	120	85	28
	CB[8] gel	90	0.0046	24	~130	--	0.75	3	~100	29
	HN-PH ₆	80.4	0.27	4.12	3.02	4.03	--	0	~85	30
	D-hydrogel-0.15	60-70	~1.75	7.48	5.9	27.2	--	240	87.6	31
	CCP-MCP1 gel	32	0.145	5.49	2.6	--	1.33	>15	--	32
Double-network hydrogel	Ca ²⁺ -alginate-PAAm	86	0.029	23	0.156	--	8.7	1440	74	14
	B-DN3 gel	44	22	5.7	10.5	--	2.85	5	~85	33
	Agar/PAMAAc-Fe ³⁺ DN	--	0.27	14	1.55	16.7	0.894	20	95	34
	DN-Sul gel	54.6	0.8	5.05	3.7	7.6	9.8	240	>90	35
	DN-Cit gel	56.9	1.3	5	5.6	12.1	14	240	96.6	35
	PAM-CS-S DN gel	~80	0.357	5.6	1.94	--	8.3	240	90	36
Nanocomposite hydrogel	L-NC gel	62	43.2	7.4	1.6	7.38	--	--	--	37
	SHARK hydrogel		0.03	77	1.02	32.6	19.75	--	--	38

(*) The concentration of acrylamide in the precursor of hydrogels was 450 mg mL⁻¹.